# NSUN2 facilitates DICER cleavage of DNA damage-associated R-loops to promote repair

Adele Alagia ®, Kamal Ajit, Arianna Di Fazio, Qilin Long & Monika Gullerova ® ✉

DNA integrity is constantly challenged by both endogenous and exogenous damaging agents, resulting in various forms of damage. Failure to repair DNA accurately leads to genomic instability, a hallmark of cancer. Distinct pathways exist to repair different types of DNA damage. Double-strand breaks (DSBs) represent a particularly severe form of damage, due to the physical separation of DNA strands. The repair of DSBs requires the activity of RNA Polymerase II (RNAPII) and the generation of Damage-responsive transcripts (DARTs). Here we show that the RNA $m^5C$-methyltransferase NSUN2 localises to DSBs in a transcription-dependent manner, where it binds to and methylates DARTs. The depletion of NSUN2 results in an accumulation of nascent primary DARTs around DSBs. Furthermore, we detect an RNA-dependent interaction between NSUN2 and DICER, which is stimulated by DNA damage. NSUN2 activity promotes DICER cleavage of DARTs-associated R-loops, which is required for efficient DNA repair. We report a role of the RNA $m^5C$-methyltransferase NSUN2 within the RNA-dependent DNA damage response, highlighting its function as a DICER chaperone for the clearance of non-canonical substrates such as DARTs, thereby contributing to genomic integrity.

Genetic information, encoded in DNA, is constantly exposed to environmental or endogenous damaging agents, leading to numerous damaged sites within the genome. To preserve DNA integrity, cells have developed proficient damage response and repair mechanisms[1]. Failure or errors in DNA repair can result in the accumulation of damage, which negatively affects cellular viability and contributes to diseases, particularly cancer[2]. Double-Strand Breaks (DSBs), caused by the simultaneous breakage of both strands of the DNA double helix, represent the most toxic form of DNA damage. DSBs are primarily repaired through the rapid and error-prone process of non-homologous end joining (NHEJ) or the more accurate homologous recombination (HR)[3].

The activation of the DNA Damage Response (DDR) involves both protein and RNA molecules. Recent evidence points to the role of various long non-coding RNAs in RNA-dependent DNA repair (RDDR), acting in *cis* or *trans*[4].

DSBs can serve as functional promoters, triggering de novo transcription that generates long non-coding transcripts crucial for efficient DNA repair. In particular, the MRN complex coordinates the preinitiation complex (PIC) formation at DSB ends, which facilitates the recruitment of active RNA Polymerase II (RNAPII) to initiate bidirectional transcription[5,6].

This process results in the generation of damage-induced long non-coding RNAs (dilncRNAs), which help to recruit various DDR factors[7]. Moreover, dilncRNAs can directly facilitate DDR by promoting end resection, a critical step in HR repair. We showed previously that c-Abl mediated phosphorylation of the Y1 CTD RNAPII, induced de novo transcription at DSBs. Upon DNA damage, Y1P RNAPII transcribes away from DSBs, producing long transcripts (primary DARTs) that hybridise with their template DNA to form R-loops. This action initiates antisense transcription directed towards the DSB (secondary DARTs), leading to the formation of double-stranded RNAs (dsRNAs).

Sir William Dunn School of Pathology, South Parks Road, Oxford, UK. ✉ e-mail: monika.gullerova@path.ox.ac.uk

DARTs are crucial in the recruitment of DDR factors, such as 53BP1 and MDC1, thereby facilitating effective DDR[8].

Interestingly, dilncRNAs may be processed into small non-coding RNAs, known as DNA damage response RNAs (DDRNAs), which have direct roles in DDR. The depletion of DICER and DROSHA, key RNA processing enzymes in RNA interference (RNAi), significantly reduces the formation of foci containing DDR factors such as 53BP1, pATM, and MDC1[9–12]. Notably, the knockdown of other factors involved in miRNA biogenesis, including the three GW182-like proteins, does not affect DDR foci formation, indicating a miRNA-independent role for DROSHA and DICER in DDR.

Post-transcriptional RNA modifications, including N6-methyladenosine (m6A), N1-methyladenosine (m1A), 5-methylcytosine (m5C), 2′-O-methylation (2′-OMe), and pseudouridine (Ψ), play crucial roles in regulating various aspects of RNA biology, including stability, splicing, localisation, and translation. These modifications are dynamically regulated by "writer", "reader", and "eraser" enzymes. Each of these enzymes contributes to the modification process, ensuring proper cellular function. Furthermore, RNA modifications have been linked to DDR pathways, highlighting their significance in the maintenance of genome integrity[13,14]. For example, m6A RNA methylation facilitates the recruitment of Pol κ to UV-induced DNA lesions, thus accelerating repair[15]. Moreover, m6A RNA modifications, mediated by METTL3 and recognized by YTHDC1, accumulate at DSBs to promote HR[16]. Likewise, m6A modifications on RNA within DNA:RNA hybrids at DSBs modulate their stability[16]. Similarly, m5C accumulates at damaged sites and contributes to DNA repair, particularly within the context of R-loops. The recruitment of the writer enzyme DNMT2 to DNA damage sites is crucial for m5C -mediated repair, as its deficiency hinders HR[17]. NSUN2, another m5C methyltransferase, is primarily known for tRNA-methylation, but has also been implicated in modifying other RNA substrates, such as mRNA and long non-coding RNA (lncRNA)[18–21]. NSUN2 amplification/overexpression is common in many cancers[22]. Therefore, the understanding of NSUN2's role in DSB repair is of great importance.

Here we show that NSUN2 is localised at DSBs in a transcription-dependent manner. Moreover, using a modified FISH-PLA (zC-FISH-PLA), zC-ChIP-seq and Direct RNA sequencing by Oxford Nanopore Technology approaches[23], we show that NSUN2 binds to and methylates DARTs. NSUN2 depletion results in an accumulation of nascent RNA around DSBs, particularly primary DARTs. Furthermore, NSUN2 interacts with DICER in RNA-dependent manner and this interaction is stimulated by DNA damage. NSUN2 activity facilitates DICER-mediated cleavage of DARTs-associated R-loops, which is essential for efficient DNA repair.

We propose a role for the m5C-methyltransferase NSUN2 within RNA-dependent DDR. Our findings provide insights into the function of NSUN2 as a DICER co-factor for the cleavage and clearance of non-canonical substrates such as DARTs.

## Results

### NSUN2 is recruited to DNA double strand breaks in a transcription-dependent manner

A previous study on scaRNA2 and DNA damage repair revealed its role in stimulating HR-mediated DNA damage response[24]. Additionally, a miCLIP assay identified scaRNA2 as a substrate for NSUN2-mediated m5C modification[25]. To investigate the role of NSUN2 in the context of DNA damage, we first performed a Proximity Ligation Assay (PLA) using antibodies against NSUN2 and γH2AX (a DNA damage marker) following ionizing irradiation (IR)-induced DNA damage in U2OS cells. Upon DNA damage induction ( + IR), we detected a significant increase in PLA signal, indicating spatial proximity between NSUN2 and γH2AX at DSBs. This interaction was abolished upon NSUN2 knockdown and was markedly reduced by transcriptional inhibition using 5,6-dichloro-1-β- d -ribofuranosylbenzimidazole (DRB) or triptolide (TPL),

suggesting that NSUN2 recruitment is transcription-dependent. Controls using single antibodies confirmed the specificity of the PLA signal (Fig. 1a). Additionally, we performed this PLA in an IR dose-dependent manner and showed that increasing number of PLA foci correlated with increased IR dose (2 Gy; 5 Gy; 10 Gy) (Supplementary Fig. 1a). Finally, to exclude the possibility of unspecific pan-nuclear PLA signal, we performed this assay using antibodies against HOXD11 (known transcription factor with no role at DSBs), and γH2AX. No significant increase in PLA foci was observed upon 10 Gy irradiation (Supplementary Fig. 1b), whilst the expression of HOXD11 was detectable in both, undamaged and irradiated cells (Supplementary Fig. 1c).

Furthermore, to assess the dynamics of NSUN2 recruitment to DNA damage sites, we employed live-cell microscopy combined with laser micro-irradiation in U2OS cells expressing NeonGreen-tagged NSUN2. Laser micro-irradiation induced localised DNA damage sites, resulting in rapid accumulation of NSUN2 at damage sites, visible as enriched fluorescence at the irradiated region within seconds post-damage (Fig. 1b). In contrast, NeonGreen alone failed to show enrichment. Time-lapse quantification confirmed specific NSUN2 accumulation at DNA damage sites over a 96-second time course.

Finally, to determine whether NSUN2 catalytic activity is required for its recruitment to DNA damage sites, we compared the dynamics of wild-type NSUN2 and the catalytically inactive NSUN2 K190M mutant using live-cell imaging and laser micro-irradiation in HEK293T. Both constructs accumulated rapidly at DNA damage sites, exhibiting comparable recruitment kinetics, indicating that RNA methylation activity is dispensable for NSUN2 localisation to DNA damage sites. However, treatment with the RNA polymerase II inhibitor TPL significantly impaired the recruitment of both NSUN2 wild-type and K190M mutant proteins (Supplementary Fig. 1d). Together, these results demonstrate that NSUN2 is rapidly recruited to DNA damage sites in a transcription-dependent but methylation-independent manner, supporting a role for NSUN2 at DNA damage sites linked to active transcriptional regions.

### NSUN2 binds to and methylates antisense DNA damage-responsive transcripts (DARTs)

Given that NSUN2 recruitment to DNA damage sites depends on active RNAPII transcription, and that NSUN2 is an RNA methyltransferase, we hypothesised that it might interact with RNA transcripts derived from DSBs. To investigate this, we employed a 5-aza-Cytosine–based crosslinking strategy in the AsiSI-ER U2OS cell line (DIvA system). 5-aza-Cytosine (5-aza-C) is a cytosine analogue that can trap RNA methyltransferases like NSUN2 by forming an irreversible covalent complex with the enzyme. NSUN2 normally catalyses the methylation of cytosine at the carbon-5 (C5) position in RNA to form 5-methylcytosine (m5C), a modification important for RNA stability, translation, and stress responses. This process involves the enzyme recognizing its RNA substrate, where a catalytic cysteine in NSUN2 performs a nucleophilic attack on the C6 position of the cytosine ring, forming a covalent intermediate. A methyl group from S-adenosylmethionine (SAM) is then transferred to the C5 position of cytosine, after which the enzyme is released. However, when 5-aza-C is incorporated into RNA in place of cytosine, its nitrogen atom at the C5 position prevents the methyl group from being transferred, blocking the completion of the methylation reaction. NSUN2 still forms the initial covalent intermediate with 5-aza-C at the C6 position, but because the methylation step cannot proceed, the enzyme becomes irreversibly trapped on the RNA. This mechanism is exploited experimentally to identify RNA targets of NSUN2 by "trapping" the enzyme at its binding sites (Fig. 2a)[25]. The DIvA system allows sequence specific induction of DSBs after the addition of (Z)−4-Hydroxytamoxifen[26].

We performed NSUN2 chromatin immunoprecipitation followed by sequencing (ChIP-seq) in AsiSI-ER U2OS cells upon no damage or DNA damage conditions (−/+4OHT) (Supplementary

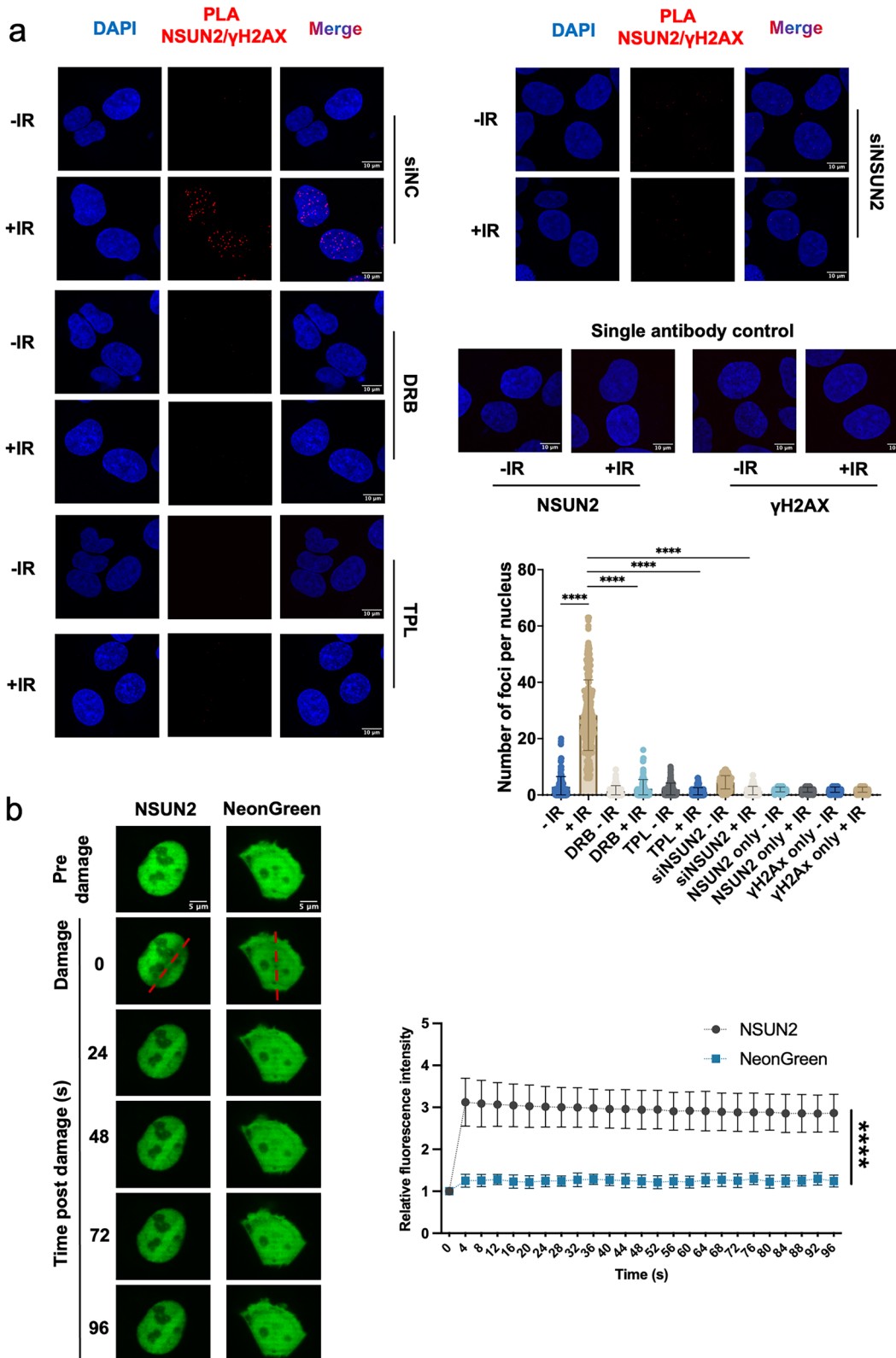

Fig. 2a). NSUN2 was significantly enriched at BLESS80 AsiSI cut sites, (a curated set of 80 high-confidence DSB loci). ChIP-seq signal peaked at these DSBs and declined with distance (Fig. 2b, c), while enrichment was markedly lower under no damage conditions (Fig. 2d, e), confirming DSB-induced recruitment of NSUN2. To further investigate the influence of genomic context on NSUN2's presence at DSBs, we examined NSUN2 enrichment at

specific AsiSI-induced DSBs categorised by transcriptional activity and repair pathway preference. We observed that NSUN2 preferentially accumulates at transcriptionally active, HR-prone, and NHEJ-prone DSBs, while showing no significant enrichment at uncut genomic regions (Supplementary Fig. 2b–g). Finally, we validated NSUN2's presence at a selected DSB (DS1; RBMXL1) by ChIP-qPCR (Supplementary Fig. 2h).

**Fig. 1 | NSUN2 is recruited to double strand breaks. a** Left and top right: Representative proximity ligation assay (PLA) images showing the interaction between NSUN2 and γH2AX in U2OS cells treated with ionizing radiation (+ IR) or untreated (−IR) (10 Gy, 15 min). PLA signal between NSUN2 and γH2AX following IR with or without transcriptional inhibitors 5,6-Dichloro-1-β-D-ribofuranosylbenzimidazole (DRB) or triptolide (TPL), indicating transcription dependency. Single antibody (NSUN2 and γH2AX) and siNSUN2-treated controls confirm PLA specificity. Cells were co-stained with DAPI. Scale bars, 10 μm. Bottom right: Scattered box plot show quantification of PLA nuclear foci. ****$p \leq 0.0001$. Non-significant comparisons ($p > 0.05$) are not shown. **b** Left: representative time-lapse images of U2OS cells before and after laser-induced DNA damage. Cells were transiently transfected with either NeonGreen-tagged NSUN2 (NSUN2) or NeonGreen-only control (NeonGreen) constructs. Localised DNA double-strand breaks were generated by targeted irradiation with a 405 nm laser across a predefined region of interest (ROI), following pre-sensitisation with 10 μM Hoechst 33342 for 30 min at 37 °C to facilitate photosensitisation. Live-cell fluorescence imaging was performed over a 96-second time course. Scale bars, 5 μm. Right: quantification of fluorescence intensity as the ratio of ROI signal to background signal over time, $n = 3$. Data are presented as mean values ± Standard Error (SEM). Statistical analysis was performed using two-way ANOVA with multiple comparisons. ****$p \leq 0.0001$. Source data are provided as Source Data file.

To strengthen our hypothesis that NSUN2 is recruited to DSBs and deposits m5C on DARTs, we exploited Nanopore Direct RNA Sequencing (DRS) in the DIvA system. DRS interrogates native chromatin-enriched RNAs (cheRNAs) extracted after site-specific AsiSI cleavage, preserving modification signatures; the distinctive ionic-current shifts caused by m5C, decoded by trained models, allowed us to pinpoint NSUN2-dependent methylation events with nucleotide resolution (Supplementary Fig. 3a).

Metagene analysis revealed a significant increase in m5C levels at NSUN2-bound regions in control cells (+4OHT siNC), compared to NSUN2 knockdown (+4OHT siNSUN2) (Fig. 2f and Supplementary Fig. 3b–d). IGV snapshots confirmed the presence of m5C signal at representative loci including PCGF1 and HEBP1, which was decreased upon NSUN2 depletion (Fig. 2g and Supplementary Fig. 4a–c).

Together, these results demonstrate that NSUN2 is recruited to transcriptionally active DSBs, where it binds to and methylates RNA species in proximity to the break sites. This suggests a role for NSUN2 in the post-transcriptional modification of DARTs during the DNA damage response.

## NSUN2 selectively binds to and methylates antisense DARTs at transcriptionally active DSBs

Following our observation of NSUN2-dependent m5C deposition on DARTs using DRS, we next aimed to validate these interactions at single-cell resolution. To visualise NSUN2–RNA interactions at specific DSBs, we developed zC-FISH-PLA, an enhanced FISH-PLA strategy that couples 5-aza-cytidine (zC)–mediated covalent trapping of NSUN2 on nascent transcripts with locus-specific DNA oligonucleotide hybridisation. Upon incorporation into nascent transcripts, zC forms an irreversible covalent adduct with the catalytic cysteine of NSUN2, thereby trapping the methyltransferase on its RNA substrate. Strand-specific probe sets spanning a 1 kb interval centred on the DS2 break target the antisense (DS2-AS) and sense (DS2-SS) damage-associated RNAs (DARTs), positioning the proximity-ligation readout over the engaged transcripts. This integrated configuration provides high-resolution, strand-resolved detection and quantitative assessment of NSUN2-bound RNA at discrete genomic lesions (Fig. 3a).

zC-FISH-PLA at the DS2 intragenic locus (VSTM2B, AsiSI site is located in the intron between exon 3 and exon 4) revealed a robust interaction between NSUN2 and antisense DARTs following DSB induction (Fig. 3b, c), whereas no signal was observed for sense strand transcripts (Supplementary Fig. 5a). Quantification of zC-FISH-PLA nuclear foci showed a significant reduction upon transcriptional inhibition with triptolide (TPL) (Fig. 3b, c) or NSUN2 depletion, demonstrating the specificity and transcription dependence of the assay (Supplementary Fig. 5b). Interestingly, zC-FISH-PLA signal was diminished by treatment with RNase H, suggesting involvement of DARTs in R-loops formation. Additionally, we treated cells with Exonuclease VII to test whether zC-FISH-PLA potentially hybridises to the DNA overhangs after resection. This treatment did not affect zC-FISH-PLA foci levels, further confirming its specificity to RNA (Fig. 3b, c). Scrambled probes did not yield any detectable signal, supporting the sequence specificity of the assay, whereas nuclear foci were still detectable in the absence of zC pre-treatment, indicating that zC incorporation enhances but is not essential for the detection of NSUN2–RNA interactions (Supplementary Fig. 5c and d). Co-localisation of zC-FISH-PLA foci with γH2AX staining further confirmed the association of NSUN2–RNA complexes with DNA damage foci (Fig. 3d). Notably, no NSUN2–RNA foci were detected when targeting DS3, a transcriptionally inactive intergenic AsiSI site, indicating that NSUN2 association is selective for active DSB loci (Supplementary Fig. 6a–b). Finally, we assessed whether this interaction with antisense DARTs was specific to NSUN2 by performing zC-FISH-PLA using DNMT2, another m5C RNA methyltransferase previously implicated in R-loop methylation following oxidative stress[17]. DNMT2 did not exhibit detectable association with either antisense or sense DARTs at the DS2 site (Supplementary Fig. 6c–d), supporting the notion that selective recognition and methylation of antisense DARTs is a unique feature of NSUN2.

These findings indicate that NSUN2 selectively associates with and methylates newly synthesized antisense DARTs at DSBs in a transcription-dependent manner, setting it apart from DNMT2, which does not exhibit similar interactions under these conditions.

Interestingly, NSUN2's strand-selective binding at intragenic DSBs is consistent with the model proposed by de Almeida and colleagues, whereby intragenic DNA damage promotes antisense transcription initiation[27].

## Depletion of NSUN2 stabilises levels of antisense DARTs

To determine whether NSUN2 modulates the stability of DARTs, we performed chromatin-associated RNA sequencing (Chr-RNA-seq) in DIvA U2OS cells under DNA damage conditions (+4OHT) following NSUN2 knockdown (siNSUN2) or control treatment (siCtrl). Principal component analysis confirmed high reproducibility across biological replicates (Supplementary Fig. 7a). Analysis of sense and antisense transcript coverage surrounding the 80 most efficiently cleaved AsiSI-induced DSBs revealed a marked accumulation of antisense DARTs in NSUN2-depleted cells compared to control, with only minor effects on sense transcripts (Fig. 4a; and Supplementary Fig. 7b).

Quantification of log2 fold changes in transcript abundance within ±500 bp of the BLESS80 sites confirmed a significant increase in antisense coverage upon NSUN2 depletion relative to sense transcripts (Fig. 4b; Supplementary Fig. 8a). As a control we tested induction of γH2AX in AsiSI-ER U2OS cells after addition of 4OHT in the presence or absence of NSUN2. Depletion of NSUN2 did not affect γH2AX induction (Supplementary Fig. 8b). DART enrichment was further visualised in metagene profiles, which demonstrated elevated antisense signal centred around DSBs in NSUN2 knockdown cells (Fig. 4c), with similar trends observed at representative loci including CREBZF and PCGF1 (Fig. 4d, e). These effects were not observed at uncut AsiSI sites, where transcript levels remained unchanged (Supplementary Fig. 7c, d).

To assess whether NSUN2-sensitive DART stabilisation is influenced by genomic context, we categorised DSBs based on transcriptional activity. NSUN2 depletion led to a robust accumulation of both sense and antisense DARTs at DSBs located in highly transcribed

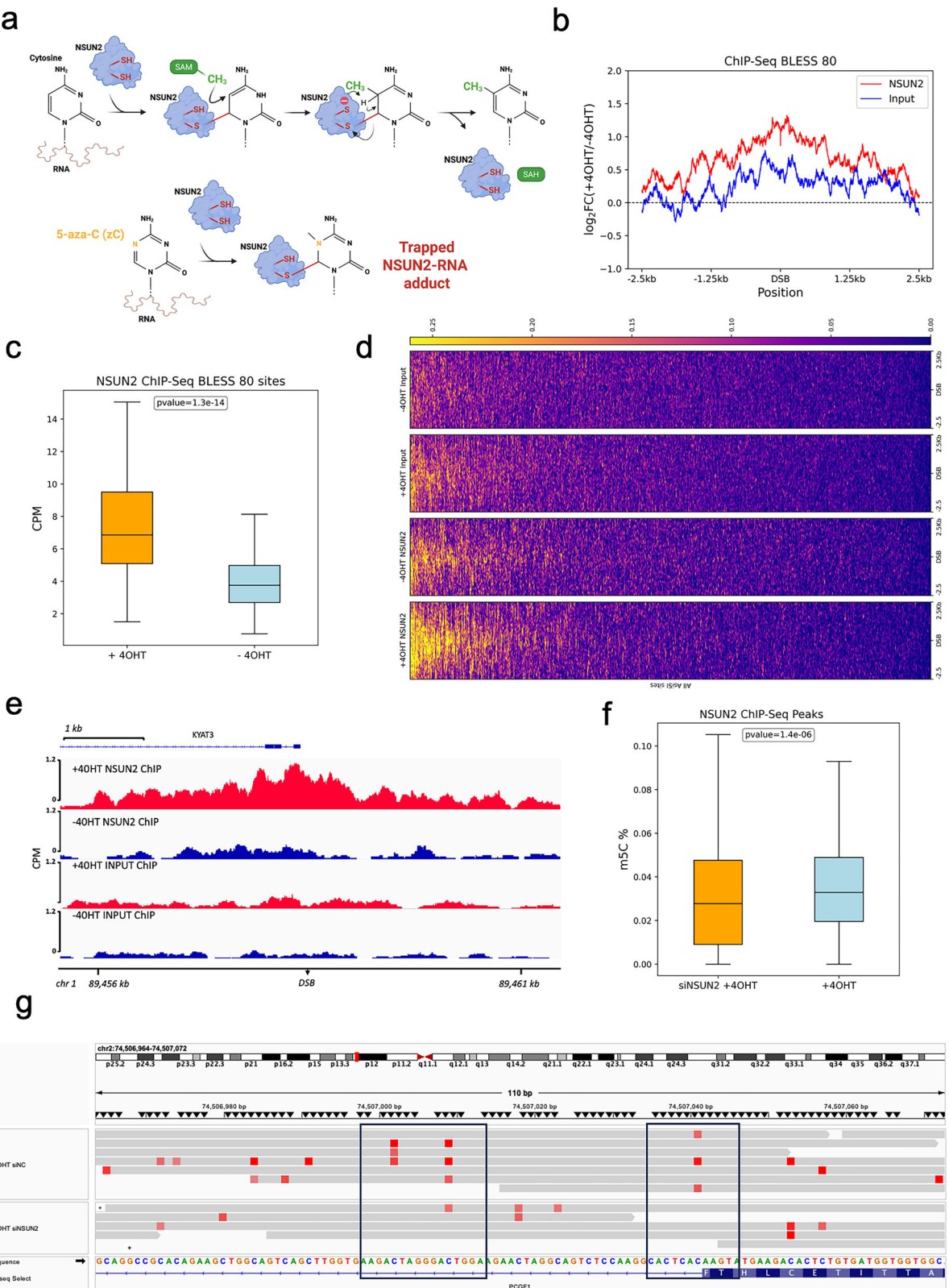

regions, while at low transcriptional activity sites, only antisense transcripts were increased (Fig. 4f; and Supplementary Fig. 8c–e). Similar patterns were observed when DSBs were further stratified by repair pathway use. At homologous recombination (HR)-prone sites (defined by high RAD51 and low XRCC4 ChIP-seq signal) NSUN2 knockdown resulted in elevated levels of antisense DARTs (Fig. 4g; and

Supplementary Fig. 8f). In contrast, at non-homologous end joining (NHEJ)-prone sites, both sense and antisense DARTs were stabilised in the absence of NSUN2 (Supplementary Fig. 8g and h). These findings identify NSUN2 as a critical regulator of DART dynamics, exerting a marked influence on antisense transcript levels at transcriptionally active and HR-prone DSBs.

**Fig. 2 | NSUN2 binds to and methylates antisense DNA Associated RNA Transcripts (DARTs). a** Schematic of the NSUN2 catalytic mechanism. NSUN2 transfers a methyl group from S-adenosylmethionine (SAM) to the C5 position of cytosine in RNA, forming 5-methylcytosine (m$^5$C) via a covalent enzyme–RNA intermediate involving a conserved cysteine residue. Following methylation, S-adenosylhomocysteine (SAH) is released. Incorporation of 5-azacytidine (5-aza-C) into RNA traps NSUN2 in a covalent complex due to a nitrogen at C5, preventing methyl transfer. This mechanism enables immunoprecipitation-based identification of NSUN2-bound RNAs. Created in BioRender. Gullerova, M. (https://BioRender.com/yf3bx22). **b** Metagene analysis showing average log$_2$ fold enrichment of NSUN2 ChIP-Seq signal (red) relative to Input (blue) across ±2.5 kb windows centred on 80 AsiSI-induced DNA double-strand break (DSB) sites (BLESS80) in DIvA U2OS cells. **c** Box plot of CPM-normalized log$_2$ NSUN2 ChIP-Seq signal at the 80 AsiSI DSBs under DNA damage (+4OHT, orange) and control (−4OHT, blue) conditions. **d** Heatmaps of NSUN2 and Input ChIP-Seq signal across ±2.5 kb surrounding the 80 DSB sites, under −4OHT and +4OHT conditions. Each row represents one DSB site, ranked by cleavage efficiency (top to bottom). Yellow indicates higher signal; dark purple indicates lower signal. **e** Snapshot of NSUN2 and Input ChIP-Seq signal at the Kyat3 locus, a representative DSB site within the BLESS80 group. Normalized tracks are shown under −/+4OHT conditions. RefSeq annotations indicate gene structure and orientation. **f** Box plot showing m$^5$C levels in chromatin-associated RNA, measured by direct RNA sequencing (DRS) within ±2.5 kb of NSUN2-bound AsiSI sites. Comparison is shown for DNA-damaged (+4OHT) cells with NSUN2 (orange) or after NSUN2 knockdown (siNSUN2, blue). **g** IGV snapshot of DRS-detected m$^5$C modifications at the PCGF1 locus in +4OHT-treated cells under control or siNSUN2 conditions. Red rectangles denote predicted m$^5$C sites (>60% probability) in antisense RNA reads. Reference nucleotide and amino acid sequences are shown for orientation. Source data are provided as Source Data file.

## DICER interacts with NSUN2 and DARTs

The accumulation of antisense DARTs upon NSUN2 depletion supports a model in which NSUN2-mediated methylation facilitates their turnover or processing at sites of DNA damage.

As NSUN2 lacks intrinsic RNA cleavage activity, we hypothesised that its role in DART processing involves cooperation with other RNA-processing factors. Given prior evidence linking DICER to DNA repair and its recruitment to DSBs, we reasoned that NSUN2 could functionally associate with DICER. To test this, we employed Surface Plasmon Resonance (SPR) a label-free method that monitors biomolecular binding in real time under near-physiological conditions. SPR analysis using purified recombinant proteins confirmed a direct NSUN2–DICER interaction in vitro with sub-micromolar affinity (Fig. 5a and Supplementary 9a–c). This interaction was further validated in cells by PLA, revealing a substantial increase in NSUN2–DICER PLA foci following ionizing radiation or 4OHT-induced DSBs. Signal specificity was confirmed by single-antibody controls, and PLA foci were diminished upon transcriptional inhibition (DRB, TPL) or RNase treatment, implicating an RNA- and transcription-dependent interaction (Fig. 5b and Supplementary Fig. 10a). To exclude the possibility of an unspecific effect, we performed control PLA using 53BP1 (DNA repair factor) and γH2AX and show that this was not affected by a treatment with RNases (Supplementary Fig. 10b). Finally, we tested the proximity of DICER to DNMT2 by PLA, but failed to detect positive foci (Supplementary Fig. 10c).

We next asked whether DICER binds to antisense DARTs at damage sites. FISH-PLA with a DS2 antisense-specific probe and DICER antibody resulted in nuclear signals upon DSB induction, which increased upon NSUN2 depletion (Fig. 5c and Supplementary Fig. 11c). This suggests that NSUN2 depletion enhances DICER's interaction with antisense DARTs, potentially due to elevated RNA substrate availability. This interaction was lost upon DICER knockdown, RNase H treatment, or the use of a scrambled probe (Supplementary Fig. 11a, b and d), confirming its specificity. Furthermore, metagene analysis comparing NSUN2 and DICER ChIP-seq signals at BLESS80 DSBs revealed coordinated enrichment around cut sites (Fig. 5d). Finally, FISH-PLA signals colocalised with γH2AX-marked DSBs, supporting the recruitment of DICER to damage-associated RNA (Fig. 5e and Supplementary Fig. 11e).

Collectively, these data indicate that NSUN2 and DICER form an RNA-dependent complex regulating DARTs and implicating DICER in their downstream processing.

## NSUN2 promotes DICER processing of DART-associated R-loops

Prompted by the observed interaction between NSUN2 and DICER and their shared enrichment at antisense DARTs, we next investigated whether NSUN2 modulates DICER-mediated processing of these transcripts. To explore this, we purified recombinant wild-type NSUN2, NSUN2-K190M (a methyltransferase-dead mutant), wild-type DICER, and a catalytically inactive DICER-DEDE variant (Supplementary Fig. 9a–c), and synthesised a panel of in vitro transcribed (IVT) RNA substrates encoding the antisense DS2 sequence. These included single-stranded RNA (ssRNA), double-stranded RNA (dsRNA), DNA:RNA hybrids, canonical R-loop structures, and R-loops bearing a 5′-tail extension (Fig. 6a; and Supplementary Fig. 12–14). In vitro cleavage assays first established that DICER efficiently processes both single-stranded RNA (ssRNA) and double-stranded RNA (dsRNA) substrates. This activity remained largely unchanged in the presence of either wild-type NSUN2 or the methyltransferase-dead NSUN2-K190M mutant, indicating that NSUN2 binding or m$^5$C deposition does not modulate DICER's catalytic efficiency on these RNA molecules (Supplementary Fig. 12a–b; and 13a–b). Supporting this, RT-qPCR analysis revealed comparable degradation kinetics across all tested conditions. In contrast, reactions containing catalytically inactive DICER-DEDE or NSUN2 alone showed minimal processing, confirming the requirement for DICER's enzymatic activity. DNA:RNA hybrids, however, were not susceptible to cleavage in any condition (Supplementary Fig. 14a).

Previous studies showed that DICER is able to process non-canonical RNA substrates such as tRNAs[28] or R-loops[29]. R-loops are known to be associated with DSBs and are formed by hybridisation of DARTs to DNA templates[8,16,30–35]. Therefore, we also performed an in vitro DICER cleavage assay using an R-loop substrate. However, DICER exhibited cleavage activity towards canonical R-loop structures, which was further enhanced in the presence of wild-type NSUN2. This stimulatory effect was attenuated when the catalytically inactive NSUN2-K190M mutant was used, suggesting that the enzymatic activity of NSUN2 contributes, at least in part, to the facilitation of DICER-mediated processing of R-loops (Supplementary Fig. 14b).

Considering that R-loops in cells would include an ssRNA tail due to nascent RNA hybridisation to the DNA template, we tested DICER's processing of the R-loop with a 5′-tail substrate. Interestingly, we observed only a weak DICER processing of the R-loop 5′-end tail substrate (Fig. 6a). However, DICER was significantly stimulated by the presence of NSUN2 wt and reduced by NSUN2 K190M within the R-loop substrate. Interestingly, cleavage at the junction between the single-stranded RNA and the R-loop body was relatively unaffected by the methylation status, suggesting that m$^5$C supports DICER access and/or activity primarily within the structured R-loop domain, potentially by modulating local RNA conformation. To further interrogate the role of 5-Cytosine methylation in DICER processing, we performed in vitro transcription of R-loop substrates using 5-methylcytidine triphosphate (m$^5$CTP), thereby introducing m$^5$C residues co-transcriptionally along the RNA backbone. These synthetically methylated R-loops exhibited markedly enhanced cleavage by DICER, even in the absence of NSUN2 protein (Fig. 6a; m5C + DICER). While these results underscore the capacity of m$^5$C to promote DICER-mediated processing, it is important to note that the density of m$^5$C incorporation via IVT likely exceed those achieved by NSUN2

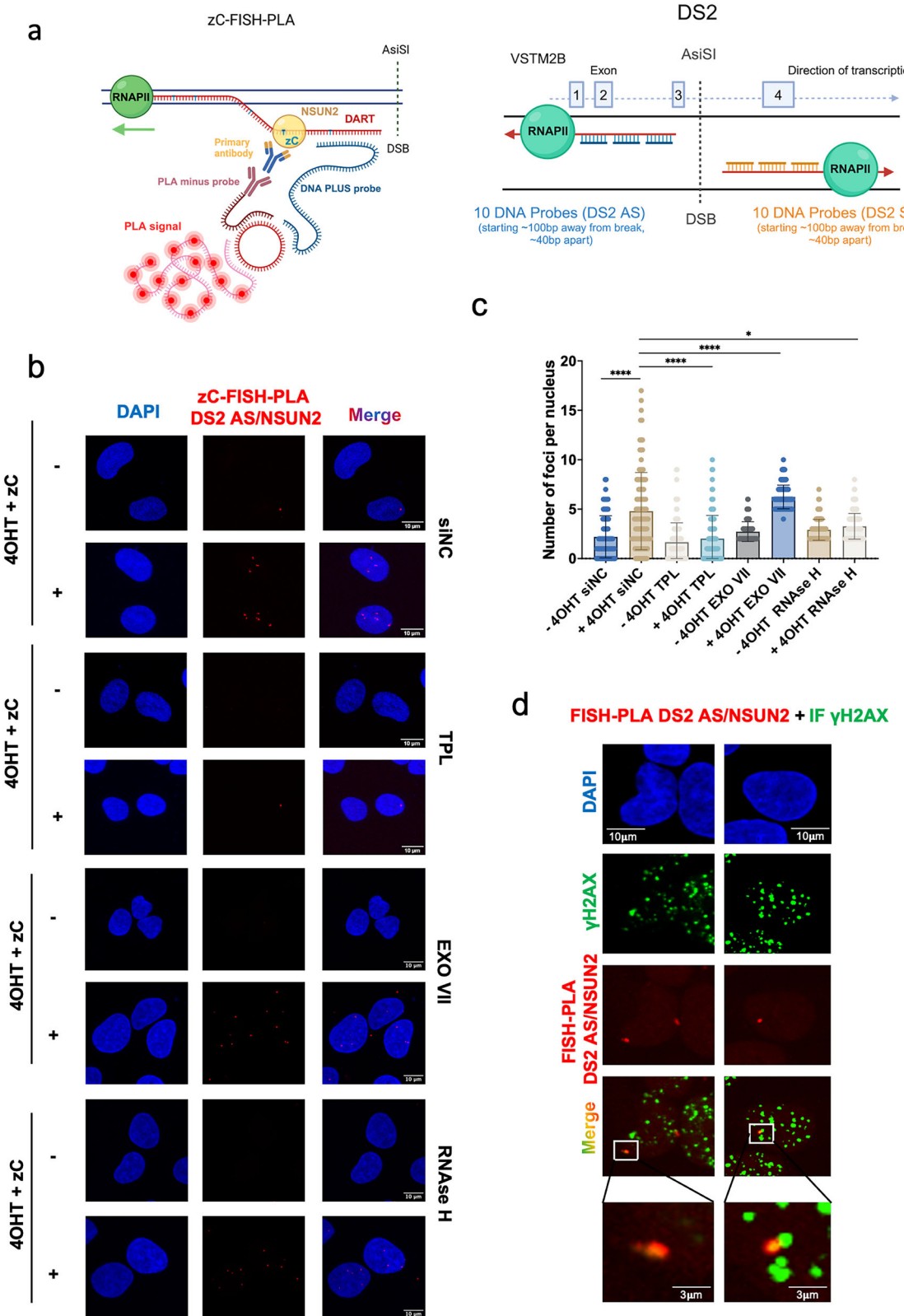

enzymatic activity in vivo. Therefore, although both methylation modalities facilitate DICER cleavage, the underlying mechanisms and biological relevance may differ, with NSUN2 potentially acting in a more targeted or structurally nuanced manner. Finally, to confirm the structural integrity of the in vitro R-loop substrates, we evaluated the formation of R-loop structures generated using unmodified RNA (RTW) and 5-methylcytosine–incorporated RNA (RT5) by slot blot

assay with the S9.6 antibody. A clear S9.6 signal was detected for both RNA species, and the signal was abolished by RNase H treatment, validating the presence of R-loop structures in each case (Supplementary Fig. 15a). These findings confirm that incorporation of $m^5C$ during IVT does not impair the ability of RNA to form R-loops.

To test the hypothesis of whether NSUN2 associates with R-loops in vivo, we performed PLA using NSUN2 and the S9.6 antibody.

**Fig. 3 | NSUN2 binds to and methylates antisense DARTs. a** Left: Schematic of the FISH–proximity ligation assay (FISH-PLA) used to detect NSUN2 interactions with damage-induced antisense RNAs (DARTs) at AsiSI-induced DSBs. Upon incorporation of 5-azacytidine (zC) into nascent RNA, NSUN2 forms a covalent RNA–protein adduct. These are detected using an anti-NSUN2 antibody and a strand-specific DNA FISH probe complementary to the target RNA. Spatial proximity allows for ligation of PLA oligonucleotides, generating a red fluorescent signal. Created in BioRender. Gullerova, M. (https://BioRender.com/) ys14435. Right: Design of strand-specific FISH probe sets targeting either antisense (DS2-AS, blue) or sense (DS2-SS, orange) DARTs at the DS2 AsiSI cut site. Each probe set includes ten oligonucleotides spaced ~40 bp apart, starting ~100 bp from the DSB. The genomic context of the DS2 site is shown with annotated *VSTM2B*, an AsiSI target gene, and RNA Pol II directionality, enabling strand-resolved detection of damage-induced RNA–protein complexes. **b** Representative FISH-PLA images showing NSUN2–RNA adduct detection at the DS2 antisense strand in DIvA U2OS cells. Cells were pre-treated with zC (1 h, 37 °C) to trap NSUN2, followed by DSB induction with (Z)–4-hydroxytamoxifen (+4OHT). PLA was performed using probes against NSUN2 and antisense DARTs. Red puncta indicate NSUN2–RNA proximity signals; nuclei were stained with DAPI (blue). Shown are untreated (−4OHT) and damaged (+4OHT) conditions, along with controls: transcription inhibition (triptolide, TPL), RNase H digestion of R-loops, and Exonuclease VII treatment to degrade 3′ ssDNA. All treatments were applied post-fixation. Images were acquired under identical settings. Scale bars, 10 µm. **c** Scattered box plot quantifying zC-FISH-PLA foci per nucleus. Statistical significance: *$p \leq 0.05$; ****$p \leq 0.0001$; ns ($p > 0.05$) not shown. **d** Representative single-plane images showing zC-FISH-PLA detection of NSUN2 and DS2 antisense RNA (red), with immunofluorescence staining for γH2AX (green). Nuclei are counterstained with DAPI (blue). Merged images show co-localisation of NSUN2–RNA complexes with DNA damage foci. White boxes indicate regions magnified in lower panels. Scale bars: overview, 10 µm; zoomed insets, 3 µm. $n = 43$ cells analysed. Source data are provided as Source Data file.

NSUN2/S9.6 proximity signals increased following DNA damage and were attenuated upon transcription inhibition (Fig. 6b), suggesting transcription-dependent association of NSUN2 with R-loops. Next, to assess the effect of NSUN2 or DICER absence on the accumulation of R-loops and methylated RNA species during DNA damage, we performed slot blot analyses. R-loop structures were detected using the S9.6 antibody, while RNA methylation was assessed using an m5C-specific antibody (Fig. 6c and Supplementary Fig.15c). As expected, IR reduced the global R-loop signal in both control and DICER-depleted cells. Notably, the depletion of NSUN2 led to enhanced R-loop signal upon DNA damage (+IR) in DICER knockdown conditions, suggesting that NSUN2 facilitates the removal or destabilisation of R-loop–containing RNAs, possibly through its functional cooperation with DICER. In contrast, DNMT2 knockdown left R-loop levels unchanged in the same experimental setting, indicating that DNMT2 is dispensable for R-loop turnover (Fig. 6c). As a control we subjected isolated R-loop structures to RNAse H treatment, which indeed confirmed the specificity of our isolation method (Supplementary Fig. 15b).

To further explore the role of RNA modifications in this context, we next interrogated global nuclear RNA methylation by slot blotting with an m5C-specific antibody (Supplementary Fig. 15c). Upon IR-induced damage, the m5C signal increased only in NSUN2-depleted cells in DICER presence, indicating that NSUN2-dependent m5C deposition is a specific mark for DICER processing. In the DICER knockdown setting, IR induced a strong m5C increase in control cells (siNC), while DNMT2 depletion (siDNMT2) resulted in elevated m5C levels regardless of damage. Interestingly, co-depletion of NSUN2 and DICER nearly abolished the m5C signal, suggesting that NSUN2-mediated m5C deposition and DICER activity function in a coordinated manner to regulate the processing of nuclear RNA molecules (Supplementary Fig. 15c). Altogether, these findings support a model in which NSUN2 deposits m5C marks on nuclear RNAs in response to DNA damage, serving as a prerequisite for DICER-mediated processing, whereas DNMT2 is dispensable for this pathway.

### NSUN2 methylation activity is crucial for efficient DNA damage repair

To examine the role of NSUN2 in DSB repair, we initially conducted comet assays in HeLa cells. In control cells, DNA damage induced by ionizing radiation (IR) was substantially repaired within 24 h. In contrast, both RAD51- and NSUN2-depleted cells exhibited sustained DNA damage at 24 h, as reflected by increased tail moments, indicating impaired DSB resolution (Fig. 7a; Supplementary Fig. 16a).

Furthermore, NSUN2-depleted cells displayed reduced viability following irradiation, with the effect becoming more pronounced over time (Fig. 7b). To assess whether the observed changes in viability were due to alterations in the cell cycle, we performed FACS-based cell cycle

analysis of HeLa cells under the following conditions: wild-type control (Ctrl), NSUN2 depletion (siNSUN2), RAD51 depletion (siRAD51), and combined depletion of both RAD51 and NSUN2 (siNSUN2 + siRAD51) (Supplementary Fig. 16b). No significant changes in cell cycle distribution were observed under the conditions tested.

To determine the specific DNA repair pathway affected, we utilised plasmid-based GFP reporters to measure homologous recombination (HR; DR-GFP) and non-homologous end joining (NHEJ; EJ5-GFP). NSUN2 knockdown alone or in combination with RAD51 depletion led to a significant decrease in HR efficiency, while NHEJ remained unaffected (Fig. 7c; and Supplementary Fig. 16c, Wortmannin treatment was used as a control).

We next addressed whether the NSUN2 protein or its catalytic activity is essential for DSB repair. Analysis of γH2AX foci revealed that NSUN2 depletion did not impair γH2AX induction shortly after IR (1 h), indicating that initial damage recognition and signalling remain intact. However, at later time points (4 and 24 h), NSUN2-depleted cells failed to efficiently clear γH2AX, pointing to a defect in the downstream repair process. This defect was rescued by wild-type NSUN2, but not by the catalytically inactive K190M mutant (Fig. 7d; and Supplementary Fig. 16d and e), emphasising the importance of NSUN2's methylation activity in promoting repair completion. FLAG-tag staining confirmed comparable expression of both wild-type and mutant constructs.

These results demonstrate that NSUN2 does not influence the initiation of DNA damage signalling, but is critical for executing homologous recombination repair through its RNA methyltransferase activity.

## Discussion

The DNA-damage response involves not only the recruitment of repair proteins, but also the tightly regulated synthesis, modification and turnover of non-coding transcripts synthesised at the site of damage. Numerous reports have demonstrated that these damage-associated transcripts are functional components of the repair process[5,7,8,27,31,32,34–51]. Immediately after DSB induction, de novo transcription is triggered at the chromatin flanking the damage site, resulting in the production of RNA transcripts called Damage-Associated RNA Transcripts (DARTs) or damage-induced long non-coding RNAs (dilncRNAs)[7,8]. These nascent RNA molecules promptly anneal to the exposed DNA ends to form break-induced RNA:DNA hybrids (BIRDHs)[50], that facilitate multiple repair steps, including recruitment of proteins, end resection and homologous strand invasion. A hallmark of these transcripts is their short-lived nature: their levels peak rapidly after break formation and are quickly processed by RNA-interfering pathway and other enzymes, making it difficult to detect under steady-state conditions[7,27,42].

The presence of de novo RNA transcripts derived from DSBs is mostly studied in systems, which involve the presence of

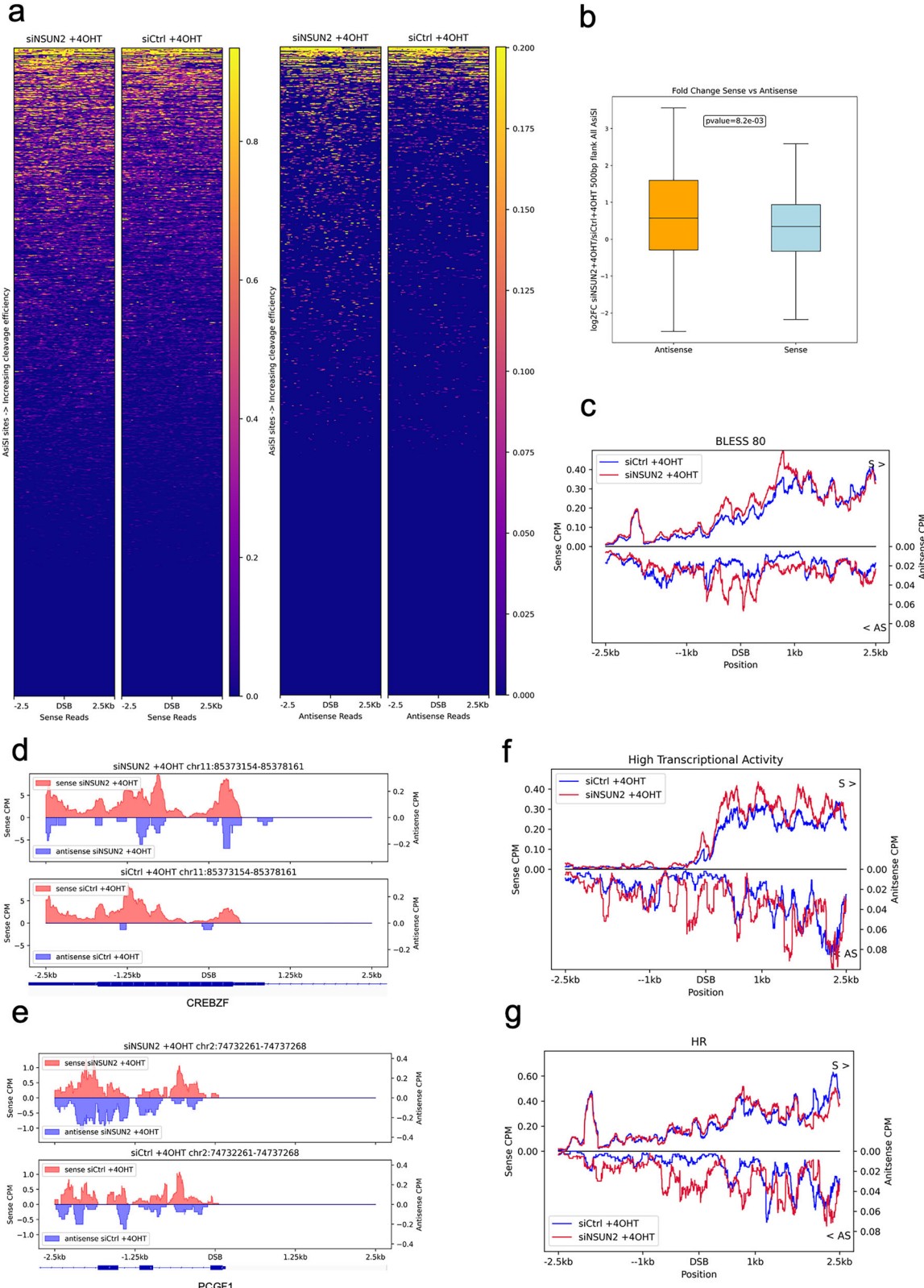

endonuclease cleavage enzyme in the nucleus. For example, the engineered reporter gene system developed by De Almeida's group[27,42], enables the DSB induction via I-SceI endonuclease within minutes. This report showed that DSBs can trigger bidirectional de novo transcription in both sense and antisense orientations and it is mutually exclusive, initiating in either the sense or antisense orientation. Furthermore, their presence depends on the genomic context,

while absent around promoter-proximal DSB, they can be significantly detected at intragenic DSB flanking region.

Interestingly, a recent genome-wide analysis from Legube and colleagues[52] failed to detect de novo transcription upon DSB induction in DIvA system, suggesting that the presence of RNA transcripts within the BIRDHs is largely subject to pre-existing RNA located at damage loci rather than to nascent RNA molecules. This however, might be

**Fig. 4 | NSUN2 depletion leads to stabilization of antisense DARTs. a** Heatmaps showing strand-specific chromatin-associated RNA sequencing (Chr-RNA-seq) read density in U2OS DIvA cells treated with (Z)−4-hydroxytamoxifen (+4OHT) to induce AsiSI-dependent DSBs. Left: sense-strand coverage in siCtrl and siNSUN2 conditions; right: corresponding antisense-strand coverage. Data are shown across ±2.5 kb flanking 80 well-characterized AsiSI cut sites (BLESS80). Each row represents one DSB site, ordered by increasing cleavage efficiency. Columns indicate genomic position relative to the DSB centre (0). Separate colour scales represent normalized signal intensity for each strand. **b** Box plot showing log$_2$ fold change (siNSUN2 / siCtrl) in Chr-RNA-seq coverage for sense and antisense strands within ±500 bp flanking BLESS80 AsiSI-induced DSBs. **c** Metagene plots of Chr-RNA-seq coverage across ±2.5 kb flanking the BLESS80 DSBs. Top: sense-strand signal; bottom: antisense-strand signal, plotted as negative values for symmetry. Read coverage is shown for siCtrl (blue) and siNSUN2 (red) conditions. Values represent average per-nucleotide coverage across all sites for each strand. **d, e** Genome browser snapshots of strand-specific Chr-RNA-seq coverage at selected AsiSI sites located near the *CREBZF* (**d**) and *PCGF1* (**e**) loci in U2OS DIvA cells. Data are shown for siCtrl and siNSUN2 conditions under +4OHT treatment. Strand-specific coverage is displayed for sense (red) and antisense (blue) transcripts. **f, g** Metagene plots of Chr-RNA-seq signal for sense (top) and antisense (bottom) strands across ±2.5 kb centred on two functional subsets of AsiSI-induced DSBs: (**f**) highly transcribed sites and (**g**) homologous recombination (HR)-prone sites. Strand-specific coverage is shown for siCtrl (blue) and siNSUN2 (red) cells after +4OHT treatment. Antisense signal is plotted as negative values to facilitate strand-resolved comparison. All data represent normalized, averaged RNA coverage per nucleotide across DSB subsets. Source data are provided as Source Data file.

reflected by technical limitations of the system used, rather than their bona fide absence. The DIvA reporter system sustains AsiSI-driven endonucleolytic cleavage of DNA for ~3–4 h, generating a heterogeneous pool of DSB lesions that simultaneously comprises newly induced DSBs, breaks already engaged in repair, and target sites excluded from repetitive cleavage due to indel formation. Such temporal asynchrony, combined with the brief 15-min 4-thiouridine pulse used for TT-seq, is therefore likely to preclude detection of the highly transient transcripts that accompany the earliest phase of DSB formation. Furthermore, the majority of cleaved AsiSI sites fall into the promoter regions, which are very well known to be GC-rich. Therefore, the incorporation of 4-thiouridine might be below the detection threshold for the TT-seq. Finally, de novo transcripts derived from DSBs, might only be detectable upon their stabilisation in loss-of-function settings such as the depletion of proteins involved in their turnover/cleavage/processing, such as INTS6, SETX, RNase H2, BRCA2, DDX17, DICER and DROSHA[7,32,34,46,47].

Our data uncover a role for the m5C methyltransferase NSUN2 as a gatekeeper of DART metabolism, thereby linking the transcriptional activity at DNA double-strand breaks to RNA modification and processing. We show that NSUN2 is rapidly recruited to DSBs, and its recruitment is transcription-dependent yet independent of its catalytic activity, positioning NSUN2 within the growing class of RNA-modifying enzymes that respond directly to nascent transcripts at DNA lesions. Importantly, NSUN2 depletion does not impair early γH2AX induction, indicating intact damage signalling, but results in delayed γH2AX resolution and defective homologous recombination repair, highlighting a specific role in repair. Genome-wide chromatin-enriched RNA-seq reveals that NSUN2 depletion leads to a marked accumulation of antisense DARTs at AsiSI break sites. The resulting accumulation of these short-lived DARTs parallels the phenotype seen after loss of important factors involved in their turnover, thereby reinforcing the principle that DARTs become detectable only when their dedicated turnover machinery is compromised[7,32,34,46,47]. Notably, the accumulation of antisense DARTs in NSUN2-deficient cells aligns with the break-proximal transcription reported by De Almeida lab, thereby strengthening the notion that de novo antisense transcription is a necessary component of the double-strand break repair programme.

Although NSUN2's catalytic activity is dispensable for its initial recruitment to DNA lesions, our data demonstrate that m5C deposition by NSUN2 is crucial for subsequent repair steps. A recent report indicated that DICER's ability to cleave R-loop structures is minimal compared to its effectiveness on hairpin RNAs when tested in vitro[29], suggesting that additional factors or RNA modifications might be required to enhance substrate recognition and processing. A key mechanistic insight from our study is the identification of a functional NSUN2–DICER complex at DSBs. Using both in vivo and in vitro approaches, we show that NSUN2 and DICER engage in a direct, RNA-dependent interaction that is enriched upon DNA damage, providing a scaffold that further promotes DICER engagement and processing of R-loop structures. Importantly, the presence of NSUN2 protein significantly enhances DICER-mediated cleavage of the 5′ tail-containing R-loops which mimic physiologically relevant nascent transcription intermediates generated at DSBs. This effect is, however, not solely linked to the direct interaction between NSUN2 and DICER. Hypermethylated 5′ tail-containing R-loop substrates are cleaved more efficiently by DICER even in the absence of NSUN2, indicating that m5C per se improves accessibility or engagement with structured RNA substrates.

In conclusion, our data reveals an additional layer of post-transcriptional regulation at DNA breaks, in which NSUN2 coordinates the methylation and turnover of DARTs through a functional partnership with DICER. This interaction facilitates the resolution of R-loops and ensures timely clearance of damage-associated transcripts, a prerequisite for effective DNA repair. NSUN2 emerges as a dual-function regulator: it acts as a damage-responsive sensor recruited to sites of active transcription, and as a catalytic effector facilitating RNA turnover through methylation-dependent stimulation of DICER activity (Fig. 8). This functional synergy between RNA modification and structured RNA resolution broadens our understanding of how transcriptome dynamics are coupled to genome stability, revealing paradigms in RNA-guided repair mechanisms.

## Methods

### Cell lines and cell culture
HeLa, HeLa EJ5-GFP, HeLa DR-GFP, HEK 293 T, HEK 293 T 2B2 (gift from the Filipowicz Lab), U2OS and AsiSI-ER (DIvA) U2OS (gift from the Legube Lab) cells were maintained under exponential growth in high-glucose DMEM medium (Life Technologies, #31966047) supplemented with 10% (vol/vol) heat inactivated fetal bovine serum (Sigma-Aldrich, #F9665), 2 mM L-glutamine (Life Technologies #25030024) and 100 units/ml penicillin-streptomycin solution (Life Technologies, #15140122) at 37 °C with 5% $CO_2$. Regular tests for mycoplasma contamination have been conducted by PCR assessment method.

The DNA damage was generated by γ-rays source.

### Plasmid construction
CDS of human NSUN2, NSUN2 K190M, DICER DEDE has been obtained from IDT as double-stranded DNA fragments (*i.e.* gBlocks HiFi Gene Fragments). Production of FLAG-NSUN2, FLAG-NSUN2-K190M, NeonGreen-NSUN2, NeonGreen-NSUN2-K190M, NeonGreen-only, pFastBac-DICER-DEDE, pFastBac-NSUN2 and pFastBac-NSUN2-K190M vectors has been obtained by seamless cloning strategy using Gibson Assembly Master Mix (NEB, #E2611S). DNA sequences of recombinant plasmids have been determined by Nanopore sequencing. A complete list of plasmids used in this study is provided in Supplementary Data 1.

### siRNA and plasmid transfection
siRNA molecules (60 nM) were reverse-transfected using Lipofectamine RNAiMax (Life technologies, #13778075) and following the

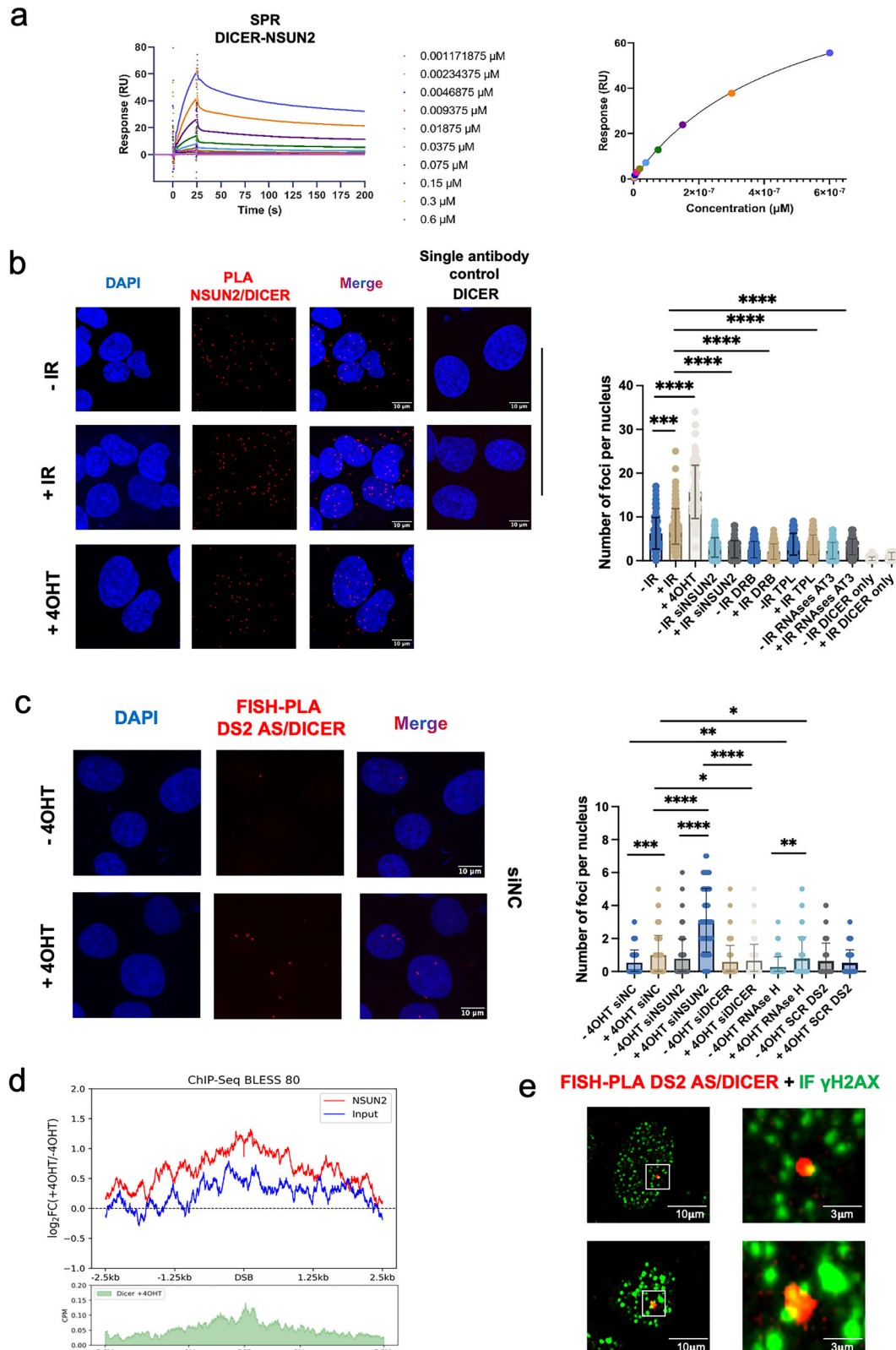

manufacturers' instructions. All experiments were conducted following a 48-hour period subsequent to siRNA transfections, unless otherwise indicated. siRNA sequences used are listed in the Supplemental information. Plasmid DNAs were forward-transfected using Lipofectamine 3000 (Invitrogen, #L3000001) and following the manufacturers' instructions. A complete list of siRNAs used in this study is provided in Supplementary Data 1.

**Proximity ligation assay (PLA) and γH2AX immunofluorescence**
PLA was performed by using Duolink In Situ Red Starter Kit Mouse/Rabbit (Merck, #DUO92101-1KT) according to the manufacturer's instructions. Cells were seeded onto poly-L-lysine coated glass coverslip to reach 60% confluency before fixing with 4% paraformaldehyde (PFA) in PBS (Alfa Aesar, # J61899) for 10 min at 37 °C. Permeabilization was performed with 0.1% Triton X-100 (Merck, #X100-100ML) in PBS

**Fig. 5 | DICER interacts with NSUN2 and DARTs. a** Surface plasmon resonance (SPR) sensorgrams (left) showing real-time binding responses of recombinant DICER (0.00117–0.6 μM) to immobilized NSUN2. Response units (RU) are plotted against time to show association and dissociation phases. Right: equilibrium binding curve plotting steady-state RU values against DICER concentration. Fitted data yield an estimated dissociation constant (K_d ≈ 0.5 μM). **b** Left: representative proximity ligation assay (PLA) images detecting NSUN2–DICER interaction in U2OS cells following DNA damage induced by ionizing radiation ( + IR) or (Z)–4-hydroxytamoxifen (+4OHT). A single-antibody control (anti-DICER) confirms signal specificity. Scale bars, 10 μm. Right: Scattered box plot quantifying PLA signals across conditions. Treatments include transcription inhibition (DRB, triptolide), and postfixation RNase digestion (RNase A, T1, III) to assess RNA dependence. Significance: ***p ≤ 0.001; ****p ≤ 0.0001; ns (p > 0.05) not shown. **c** Left: representative images showing FISH-PLA detection of DICER proximity to DS2 antisense RNA in DIvA

U2OS cells after +4OHT-induced DSBs. Scale bars, 10 μm. Right: Quantification of PLA foci per nucleus under multiple conditions: untreated or +4OHT-treated cells, NSUN2 or DICER knockdown (siNSUN2, siDICER), scrambled probe control (SCR DS2), and RNase H treatment to degrade R-loops. Significance: *p ≤ 0.05; **p ≤ 0.01; ***p ≤ 0.001; ****p ≤ 0.0001; ns not shown. **d** Metagene plots showing ChIP enrichment for NSUN2 (top) and DICER (bottom) in AsiSI-ER U2OS cells treated with +4OHT. Signals are plotted as normalized read coverage across ±2.5 kb surrounding 80 AsiSI-induced DSBs (BLESS80), centred at the cut site (0). NSUN2 signal (red) is compared to Input (blue); DICER signal is shown in green. **e** Single-plane images showing FISH-PLA signal (red) for DICER–DS2 antisense RNA complexes co-stained with γH2AX (green) after +4OHT treatment. Merged images highlight co-localisation at DNA damage foci. White boxes mark zoomed-in regions (right panels). Scale bars: overview, 10 μm; zoom, 3 μm. n = 52 cells analysed. Source data are provided as Source Data file.

for 10 min at room temperature before blocking 1 h at 37 °C with Duolink Blocking buffer. Primary antibodies were diluted (1:1000) with Duolink Dilution buffer and incubated overnight at 4 °C. Subsequent PLA probe incubation, ligation and amplification steps were performed following the manufacturer's instructions. Then coverslips were mounted with DAPI solution (Merck, #DUO82040) and sealed with transparent nail varnish before imaging with Olympus FluoView Spectral FV1200 confocal microscope with 60X oil immersion objective. Images were processed with ImageJ software and quantified by using CellProfiler with Speckle Counting pipeline. For experiments combining PLA with γH2AX immunofluorescence, cells were incubated with anti-γH2AX antibody conjugated to FITC (Miltenyi Biotec, #130-118-339; 1:1000 dilution) for 1 h at room temperature in the dark immediately following the FISH-PLA protocol. After incubation, cells were washed three times with TBS + 0.1% Tween-20 and counterstained with DAPI before imaging. A complete list of antibodies used in this study is provided in Supplementary Data 1.

### 5-aza-ChIP-seq
U2OS and AsiSI-ER U2OS cells were plated in 15 cm dishes at densities of 4.5 million cells in complete DMEM, supplemented with 5 μM 5-azacytidine and incubated for 12 h. The following day, DNA double-strand breaks (DSBs) were induced using AsiSI by adding 4-hydroxytamoxifen to complete DMEM achieving a final concentration of 500 nM. The cells were incubated for 4 h at 37 °C. Ten min before the end of DSBs induction, DSP (dithiobis[succinimidyl propionate]) was freshly dissolved in DMSO at concentration of 100 mM and then diluting dropwise into 150 ml of pre-warmed PBS containing $Ca^{2+}/Mg^{2+}$ while stirring at 37 °C, to a final concentration of 1 mM. After DSB induction, cells were washed once with PBS containing $Ca^{2+}/Mg^{2+}$, then incubated with DSP solution for 30 min at 37 °C. Subsequently, formaldehyde was added directly to the media (final concentration 1%) and incubated for 15 min at 37 °C. Crosslinking was quenched with Glycine (final concentration 125 mM) for 15 min at room temperature. Cells were washed twice with ice-cold PBS without $Ca^{2+}/Mg^{2+}$, collected by scraping using ice-cold PBS containing 0.5% BSA and pelleted by centrifugation at 500 g for 5 min at 4 °C. Then, pellets were gently resuspended in ice-cold Cell Lysis Buffer (CLB) (85 mM KCl, 5 mM PIPES, 0.5% NP-40, 1× Halt protease inhibitor cocktail (PI), and 1 mM PMSF in DEPC-treated water), prepared immediately prior to use. Cells were incubated on ice for 10 min, pelleted by centrifugation at 500 g for 5 min at 4 °C, and then resuspended in Nuclear Lysis Buffer (NLB) (50 mM Tris-HCl pH 8.0, 10 mM EDTA, 1% SDS, 1× Halt PI, and 1 mM PMSF, in DEPC-treated water). Nuclei were incubated at room temperature for 10 min prior to sonication. Chromatin was sheared using a Diagenode PICO Bioruptor at ultra-high power, with 30-second ON/OFF cycles for 30 min. The sonicated lysate was centrifuged on a tabletop centrifuge at 17,200 RCF for 10 min at 4 °C to pellet insoluble material. The soluble chromatin was pre-cleared by incubating with 1:1

mixture of Protein A and Protein G magnetic beads in ChIP IP buffer (16.7 mM Tris-HCl pH 8.0, 1.2 mM EDTA, 167 mM NaCl, 0.01% SDS, 1.1% Triton X-100, 1× Halt PI, and 1 mM PMSF in DEPC-treated water) and incubated for 1 hour at 4 °C with rotation. Each sample was then split into input controls, antibody samples, no-antibody control. The antibody samples received 5 μg of NSUN2 mouse monoclonal antibody (1 μg/μl), then were incubated overnight at 4 °C with rotation at 16 rpm. On the following day, antibody-chromatin complexes were captured by conjugation to freshly washed Protein A/G magnetic beads. Beads were washed twice in ChIP IP buffer containing protease inhibitors and subsequently added to each sample and incubated for 2 hours at 4 °C with rotation at 16 rpm. Beads were then subjected to a sequential wash with ChIP wash buffers A-D, each prepared fresh in DEPC-treated water and pre-chilled. Buffer A (20 mM Tris-HCl pH 8.0, 2 mM EDTA, 150 mM NaCl, 0.1% SDS, and 1% Triton X-1000. Buffer B (500 mM NaCl 20 mM Tris-HCl pH 8.0, 2 mM EDTA, 150 mM NaCl, 0.1% SDS, and 1% Triton X-1000). Buffer C (10 mM Tris-HCl pH 8.0, 2 mM EDTA, 250 mM LiCl, 1% sodium deoxycholate, and 1% NP-400. Buffer D (10 mM Tris-HCl pH 8.0 and 1 mM EDTA). Each wash was performed for 5 min at 4 °C with rotation, followed by 3 min on a magnetic rack. Buffers A–C were used once, and buffer D was applied twice, transferring beads to fresh tubes during the final wash. Elution was carried out by incubating beads twice with Elution Buffer (100 mM $NaHCO_3$, 1% SDS in DEPC-treated water) at 65 °C for 15 min on thermomixer with shaking at 1200 rpm, pooling eluates in fresh tubes. To perform digestion and de-crosslinking, Digestion Buffer (10×) containing 400 mM Tris-HCl pH 6.5 and 100 mM EDTA was added. RNase A (50 μg final concentration) was added to all samples and incubated at 60 °C for 3 h on thermomixer at 400 rpm. Afterward, NaCl and DTT were added to final concentrations of 300 mM and 5 mM, respectively. Proteinase K (50 μg final concentration) was then added, and samples were incubated overnight at 65 °C on thermomixer with shaking at 400 rpm. The following day, DNA was purified via phenol/chloroform extraction. DNA precipitation was performed adding Isopropanol (0.8 volumes), NaCl (300 mM final concentration), $MgCl_2$ (10 mM final concentration), and GlycoBlue (30 μg final concentration), and samples were incubated for 2 h at −20 °C. DNA was pelleted by centrifugation on a tabletop centrifuge at 17,200 RFC for 30 min at 4 °C. Pellets were washed twice with 100% ethanol and 70% ethanol. Then, pellets were air-dried for 2 h and resuspended in 50 μl of TE buffer with low EDTA (10 mM Tris-HCl pH 8.0, 0.1 mM EDTA in DEPC-treated water). DNA was quantified using the Qubit dsDNA High Sensitivity assay (ThermoFisher, #Q32851).

### FISH-PLA and γH2AX immunofluorescence
FISH-PLA and zC-FISH-PLA assays were performed as previously described (Alagia et al., 2023), with minor modifications. For zC-FISH-PLA experiments, cells were pre-incubated with 5 μM 5-Azacytidine (zC; Sigma-Aldrich, #A2385) for 1 h at 37 °C prior to

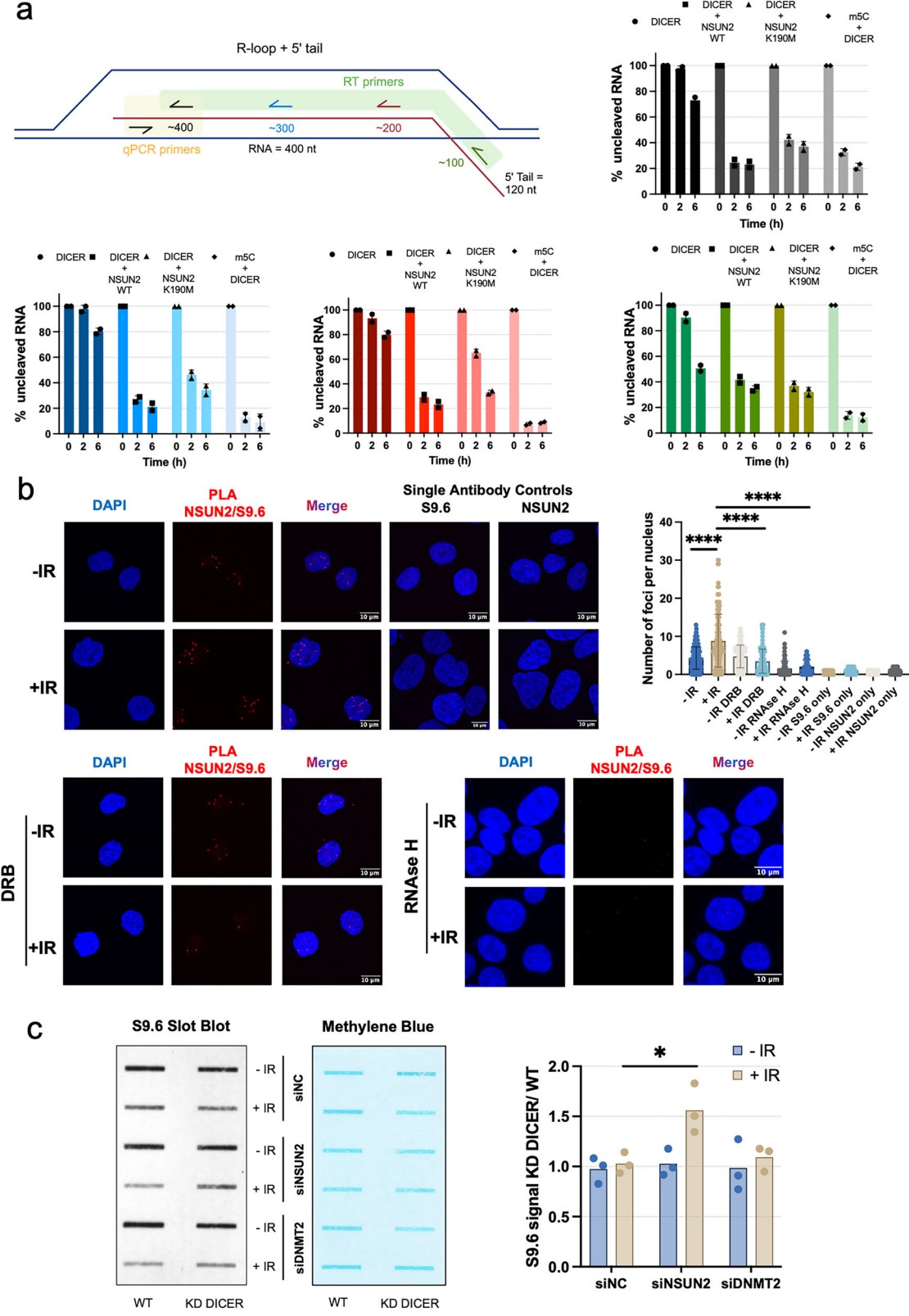

treatment with 400 nM (Z)−4-Hydroxytamoxifen (4OHT; Sigma-Aldrich, #H7904) for 4 h. For experiments combining FISH-PLA with γH2AX immunofluorescence, cells were incubated with anti-γH2AX antibody conjugated to FITC (Miltenyi Biotec, #130-118-339; 1:1000 dilution) for 1 h at room temperature in the dark immediately following the FISH-PLA protocol. After incubation, cells were washed and counterstained with DAPI before imaging.

A complete list of DNA probes and antibodies used in this study is provided in Supplementary Data 1.

**Surface plasmon resonance (SPR)**

SPR was performed using Biacore S200. Purified 6xHis-NSUN2, at 0.2 μg/mL, was immobilised on CM5 chip at pH 4.0 by amine coupling. pH (4.0–5.5) was optimised for best immobilisation conditions

**Fig. 6 | NSUN2 promotes DICER processing of DART-associated R-loops. a** Top: Schematic of the in vitro transcribed (IVT) DS2 antisense RNA construct forming an R-loop with a single-stranded 5′ tail. Four reverse transcription (RT) primers (green, red, blue, black) are positioned ~100–400 nt upstream of the 5′ tail. qPCR primers (black arrows, highlighted in yellow) target cleavage products. Bottom: Quantification of DICER-dependent cleavage at 0, 2, and 6 h in reactions containing DICER alone, DICER + wild-type NSUN2 (NSUN2 wt), or DICER + catalytically inactive NSUN2 (K190M). Both unmodified and m⁵C-modified RNA substrates were used. RT-qPCR was performed with the indicated primer sets to assess cleavage efficiency. Data are shown as mean ± SD (*n* = 2). Created in BioRender. Gullerova, M. (https://BioRender.com/ gb9kub6). **b** Left: Representative confocal images of U2OS cells showing PLA signal (red) for NSUN2–S9.6 proximity, indicating NSUN2 association with RNA:DNA hybrids. DAPI (blue) marks nuclei. PLA was performed in untreated cells (−IR), after ionizing radiation (+IR, 10 Gy, 15 min), with transcription inhibition (DRB), and following post-fixation RNase H digestion. Single-antibody controls (S9.6 only or NSUN2 only) confirmed signal specificity. Scale bars, 10 μm. Right: Quantification of nuclear PLA foci under the indicated conditions. Statistical significance: ****$p \leq 0.0001$; ns ($p > 0.05$) not shown. **c** Left: Representative S9.6 slot blot showing global R-loop levels in HEK293T and HEK 2B2 cells transfected with siRNAs targeting NSUN2, DNMT2, or a negative control (siNC), in wild-type (WT) or DICER knockdown (KD) conditions. DICER KD was achieved by doxycycline-induced shRNA expression for 3 days prior to analysis. Cells were left untreated (−IR) or irradiated (+IR, 5 Gy) and harvested after 5 min. Methylene blue staining served as a loading control. Right: Quantification of S9.6 signal intensity under DICER KD conditions, normalized to WT controls. Statistical significance: *$p \leq 0.05$; ns not shown. Source data are provided as Source Data file.

beforehand. Affinity assays were run with 2-fold serial dilutions of purified 6xHis-DICER in SPR running buffer (50 mM TrisHCl (pH 8), 50 mM NaCl, 5 mM MgCl2, 2 mM TCEP, 0.05% Tween20).

## ChrRNA-seq sample preparation
The AsiSI-ER-U2OS cells at 70% confluency were incubated with 400 nM 4-OHT for 4 h. Then, the chromatin RNA samples for sequencing were prepared according to previous publications[28,53]. RNA was extracted from the chromatin samples using TRIzol LS (Invitrogen # 10296028) and 1-Bromo-3-chloropropane (Sigma-Aldrich # B9673) extraction, followed by Monarch Total RNA Miniprep Kit (NEB) according to the manufacturer's instructions, and eluted in DEPC treated H₂O.

## Direct RNA sequencing using oxford nanopore technologies' direct RNA sequencing
Chromatin-associated RNA (Chr-RNA) was isolated and subjected to ribosomal RNA depletion using the RiboMinus Eukaryote Kit v2 (Ambion, #A15020), following the manufacturer's instructions. The resulting RNA was polyadenylated using *E. coli* poly(A) polymerase (New England Biolabs, #M0276L). Each polyadenylation reaction contained RNA template in DEPC-treated water, 1× *E. coli* Poly(A) Polymerase Reaction Buffer, 1 mM ATP, and 20 units of *E. coli* poly(A) polymerase in a final volume of 20 μL. Reactions were incubated at 37 °C for 1 min and terminated by the addition of EDTA to a final concentration of 10 mM. Polyadenylated RNA was then purified using a 1.8× volume of Agencourt RNAClean XP beads (Beckman Coulter, #A63987), incubated at room temperature for 5 min, and placed on a magnetic rack until the supernatant cleared. Beads were washed twice with 80% ethanol, air-dried for 1 min, and eluted in DEPC-treated water. RNA concentration was quantified using the Qubit RNA HS Assay Kit (ThermoFisher, #Q32851). For the following steps, components of the Direct RNA Sequencing Kit XL (SQK-RNA004-XL) were used, with minor modifications to the manufacturer's protocol.

For adapter ligation, polyadenylated RNA samples containing more than 400 ng of RNA were combined with 1× NEBNext Quick Ligation Reaction Buffer (New England Biolabs, #B6058), 1 unit/μL RNasin Plus Ribonuclease Inhibitor (Promega, #N2611), 2 μL of RT Adapter, and 2 μL of T4 DNA Ligase (New England Biolabs, #M0202M). The reaction mixture was incubated for 20 min at 23 °C in a thermal cycler.

Reverse transcription was initiated by preparing a master mix containing 2 μL of 10 mM dNTPs (New England Biolabs, #N0447), 1 × Induro Reverse Transcriptase Reaction Buffer and, 2 μL of Induro Reverse Transcriptase (New England Biolabs, #M0681). This mix was added to the adapter-ligated RNA samples and incubated at 60 °C for 30 min, followed by 70 °C for 10 min, and cooled to 4 °C.

Following reverse transcription, 2 × volume of Agencourt RNAClean XP beads were added to each sample. The mixtures were incubated for 5 min at room temperature and placed on a magnetic rack. After discarding the supernatant, the beads were washed twice with 80% ethanol, air-dried for 1 min, and RNA was eluted in DEPC-treated water.

Sequencing adapter ligation was performed by adding 1× NEBNext Quick Ligation Reaction Buffer, 3 μL of RNA Ligation Adapter, and 3 μL of T4 DNA Ligase to each eluted RT-RNA sample. Reactions were incubated for 20 minutes at 23 °C in a thermal cycler. Subsequently, 0.4 × volume of Agencourt RNAClean XP beads was added to each sample and incubated for 5 min at room temperature. Beads were washed twice with Wash Buffer (WSB), air-dried, and resuspended in RNA Elution Buffer (REB). After a 10-min incubation at room temperature, samples were placed on a magnetic rack and the eluate was recovered. The concentration of the final library was quantified using the Qubit dsDNA HS Assay Kit (ThermoFisher, #Q32851).

MinION flow cells (Oxford Nanopore Technologies, FLO-MIN004RA) were primed with a solution composed of RNA Flush Tether (RFT) and Flow Cell Flush (FCF). Sequencing libraries were prepared by mixing Sequencing Buffer (SB), Library Solution (LIS), and the RNA library eluate. Libraries were loaded onto the flow cell via the SpotON port and sequenced immediately using MinKNOW software with real-time basecalling enabled.

## Protein purification
Recombinant NSUN2, NSUN2 K190M, DICER and DICER DEDE were produced according to previous publication (Alagia et al., 2023). Briefly, Spodoptera frugiperda (Sf9) cells were infected with P1 baculovirus at MOI = 1, generated with either pFastBac1-NSUN2, or pFastBac1-NSUN2 K190M, or pFastBac1-hDICER (Addgene, #89144), or pFastBac1-hDICER-DEDE and the Bac-to-Bac expression system (Gibco # 10359016). All steps of purification were carried out at 4 °C. Cells were harvested 72 h post-infection and the lysate was loaded on pre-equilibrated Ni-NTA agarose (QIAGEN #30210). 6xHis-NSUN2, 6xHis-NSUN2-K190M, 6xHis-DICER, 6xHis-DICER-DEDE were then eluted with a gradient of Imidazole. Fractions were run on SDS−PAGE and stained with Coomassie blue. Active fractions were concentrated, and buffer was exchanged with a 100 kDa cut-off for DICER and 50 kDa for NSUN2 concentrator (Cytiva Vivaspin) and stored in storage buffer (50 mM Tris−HCl, pH 8, 50 mM NaCl, 5 mM MgCl2, 0.2% [mg/ml] BSA, 2 mM TCEP, 20% glycerol, and 1x protease inhibitor) at −80 °C.

## Synthesis of DS2-derived fragments for R-loop and DNA:RNA heteroduplex formation
Genomic DNA from U2OS cells was isolated with Phenol/chloroform/isoamyl alcohol (25:24:1, v/v) (Invitrogen #15593031), precipitated with isopropanol and resuspended in water. DsDNA for R-loop (1 kb) was PCR synthesised using primers listed in Supplemental information. ssDNA strands for DNA:RNA heteroduplex and R-loop (400nt) formation were obtained from genomic DNA using phosphorothioate and 5′-end phosphorylated primers (see Supplementary information) followed by Lambda exonuclease

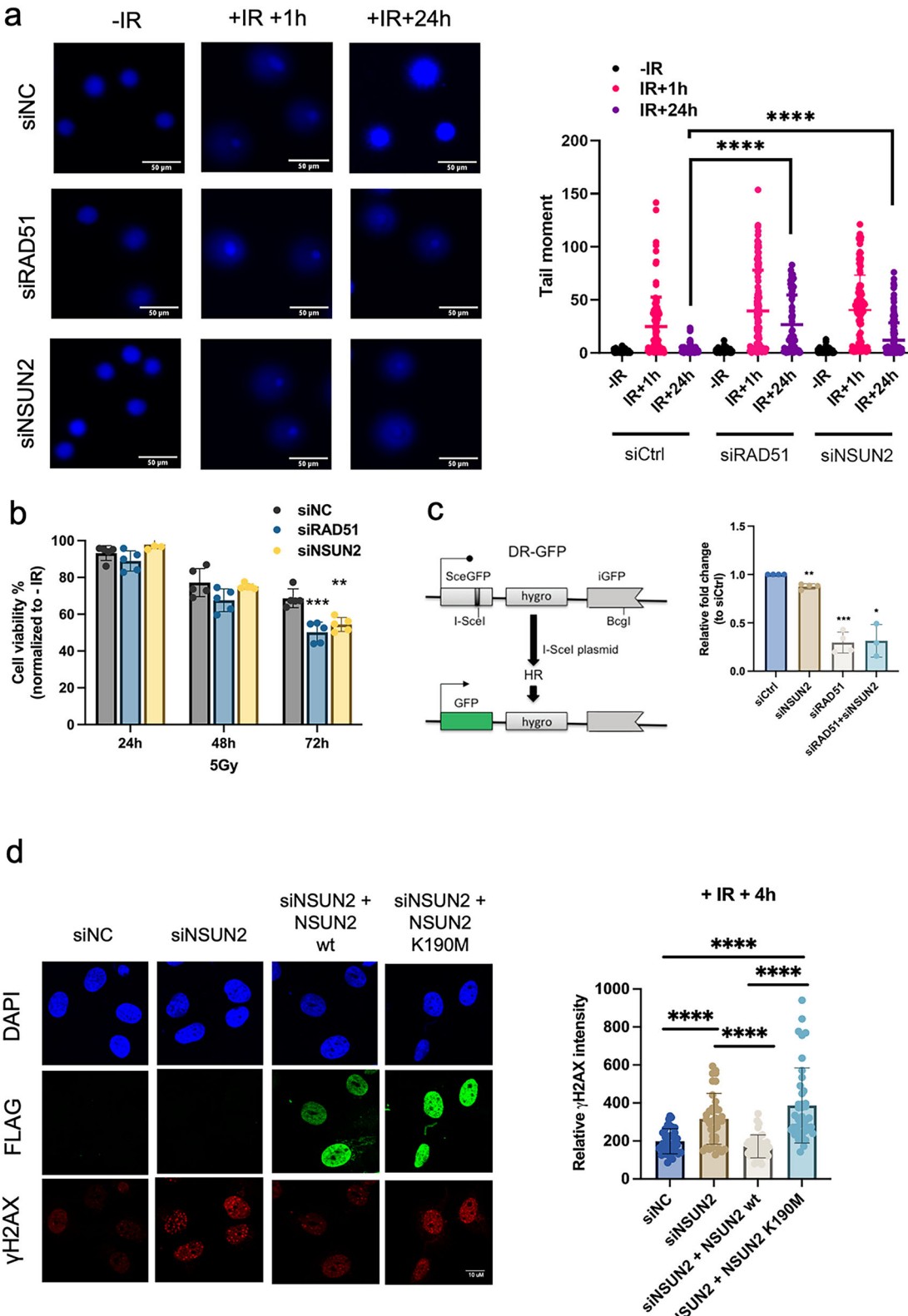

treatment (NEB # M0262S) following manufacturer's instructions. Length of DNA Fragments were determined by agarose gel electrophoresis and then purified with Monarch DNA Gel Extraction Kit (NEB # T1020L). DNA concentration was assessed by Nanodrop. DNA:RNA heteroduplexes were formed by annealing a sense single-stranded DNA (ssDNA) with antisense single-stranded RNA (ssRNA) at 98 °C for 3 min, then gradually cooled to 37 °C to promote hybridization. For R-loop formation, the sense ssDNA and antisense ssRNA strands were first annealed under the same conditions (98 °C for 3 min, slow cooling to 37 °C), followed by the addition of a complementary antisense ssDNA strand. This mixture was incubated at 70 °C for 3 min and allowed to cool slowly to room temperature to facilitate R-loop assembly. R-loop formation was validated by S9.6 slot blot assay and RNase H

**Fig. 7 | Depletion of NSUN2 delays DNA repair by HR. a** Left: Representative Comet assay images of U2OS cells treated with ionizing radiation (IR, 5 Gy, 15 min) and harvested at 1 h (+IR 1 h) or 24 h (+IR 24 h), or left untreated (−IR). Cells were transfected with siRNAs targeting NSUN2 (siNSUN2), RAD51 (siRAD51), or a negative control (siNC). Right: Quantification of DNA damage by tail moment. Data represent mean ± SD (n = 3). Statistical analysis was performed using a two-tailed Mann−Whitney test. Significance: ****$p ≤ 0.0001$; ns ($p > 0.05$) not shown. **b** MTS assay measuring cell viability in HeLa cells transfected with siNSUN2, siRAD51, or siNC, and irradiated with 5 Gy. Cell viability was assessed at 24 h, 48 h, and 72 h post-irradiation. Data are shown as mean ± SD ($n = 5$). Statistical significance was determined by two-tailed Mann−Whitney test. **$p ≤ 0.01$; ***$p ≤ 0.001$; ns not shown. **c** Schematic of the DR-GFP reporter system used to assess homologous

recombination (HR) repair efficiency. Quantification of HR activity is shown in U2OS cells transfected with siNSUN2, siRAD51, both siRNAs (siRAD51 + siNSUN2), or siNC. Statistical significance: *$p ≤ 0.05$; **$p ≤ 0.01$; ***$p ≤ 0.001$; ns not shown. **d** Left: Immunofluorescence images showing γH2AX foci formation in U2OS cells under the following conditions: siNC, siNSUN2, NSUN2 knockdown rescued with FLAG-tagged NSUN2 wild-type (siNSUN2 + NSUN2 wt) or methyltransferase-dead mutant (siNSUN2 + NSUN2 K190M). Cells were irradiated with 5 Gy and fixed at 4 h post-IR. γH2AX staining (green) marks DNA damage; FLAG-NSUN2 variants were used to assess rescue efficiency. Scale bars, 10 µm. Right: Quantification of γH2AX signal intensity per nucleus. Significance: ****$p ≤ 0.0001$; ns not shown. Source data are provided as Source Data file.

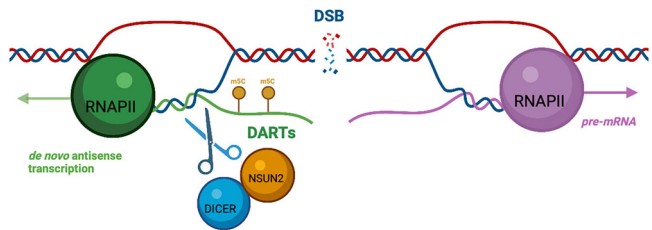

**Fig. 8 | NSUN2 promotes RNA turnover at DNA breaks via DICER-dependent degradation of DARTs.** Upon DNA damage, NSUN2 is recruited to transcriptionally active break sites, where it methylates DARTs. This modification enhances their processing by DICER, promoting RNA turnover. The NSUN2/DICER interaction facilitates R-loop resolution and ensures the clearance of damage-associated transcripts, a critical step for efficient DNA repair. Together, this synergy between RNA modification and structured RNA processing links transcriptome dynamics to genome stability. Created in BioRender. Gullerova, M. (https://BioRender.com/covu936).

digestion. A complete list of primers used in this study is provided in Supplementary Data 1.

### In vitro transcribed RNA

DICER RNA substrates were obtained using PCR synthesised linear DNA bearing T7 promoter and HiScribe T7 High Yield RNA Synthesis Kit (NEB #E2040S) following manufacturers' instructions. To achieve m⁵C modified RNA strand m⁵CTP (Jena Bioscience #NU-1138S) was added to the NTPs mixture at 1:4 ratio to CTP. After overnight reaction at 37 °C and 4U Turbo DNAse treatment for 1 h at 37 °C (Invitrogen #AM2238), RNA molecules were purified by TRIzol LS reagent and Isopropanol precipitation. RNA was resuspended in DEPC-treated H₂O and stored at −80 °C. A complete list of primers used in this study is provided in Supplementary Data 1.

### In vitro DICER cleavage assay

ssRNA, dsRNA, DNA:RNA heteroduplex, R-loop and R-loop 5'-tail substrates were pre-incubated with either 1 µg of recombinant NSUN2 or NSUN2 K190M in DICER reaction buffer (50 mM Tris−HCl, pH 8, 300 mM NaCl, 20 mM Hepes pH7.4, 5 mM MgCl₂, 5 mM CaCl₂, 0.5 mM DTT, 5% glycerol, and 1x protease inhibitor) supplemented with 0.32 mM SAM for 1 h at 37 °C, then 1 µg of either recombinant DICER or DICER DEDE were added to the reaction and incubated for 6 h at 37 °C. Aliquots were collected at certain time points (0, 2 h, 6 h) and treated with 4U Turbo DNAse for 1 h at 37 °C followed by 0.8 U Proteinase K for 1 h at at 37 °C prior isopropanol precipitation and resuspension in DEPC-treated H₂O. Then RNAs were reverse transcribed using several RT primers and SuperScript III Reverse Trascriptase (Invitrogen # 18080093) following manufacturers' instruction for gene specific primer. Before Sybr Green (Bioline # QT650-05) real time qPCR, samples have been incubated with 5U of RNAse H (NEB #

M0297S) for 1 h at 37 °C. A complete list of primers used in this study is provided in Supplementary Data 1

### Northern blot

T7 in vitro transcribed (IVT) RNA (DS2_AS) was dephosphorylated with Quick CIP (NEB # M0525S) for 30 min at 37 °C, followed by heat-inactivation at 80 °C for 2 min. Then, RNA was 5'-labelled with 20 U of T4 Polynucleotide Kinase (3´ phosphatase minus) (NEB # M0236S) and 50 pmol of (gamma-³²P)ATP (Perkin Elmer) for 1 h at 37 °C. Radiolabelled DS2_AS and either ssRNA (DS2_SS) or ssDNA were annealed by boiling at 98 °C for 3 min and then slowly cool down to room temperature over a period of no less than 3 h. ssRNA, dsRNA, DNA:RNA heteroduplex, were pre-incubated with either 1 µg of recombinant NSUN2 or NSUN2 K190M in DICER reaction buffer (50 mM Tris−HCl, pH 8, 300 mM NaCl, 20 mM Hepes pH 7.4, 5 mM MgCl₂, 5 mM CaCl₂, 0.5 mM DTT, 5% glycerol, and 1x protease inhibitor) supplemented with 0.32 mM SAM for 1 h at 37 °C, then 1 µg of either recombinant DICER or DICER DEDE were added to the reaction and incubated for 6 h at 37 °C. Aliquots were collected at certain time points (0, 2 h, 6 h), mixed with Novex TBE−urea loading dye (Invitrogen #LC6876) and immediately frozen on dry ice. Samples were run on a gel containing 7 M Urea and 4% acrylamide/bis-acrylamide (29:1) in 1x TBE at 450 V for 1 h. Then, the gel was dried, and the radiolabelled RNA was visualized by autoradiography. A complete list of probes used in this study is provided in Supplementary Data 1.

### Nuclear RNA isolation and detection of 5-methylcytosine (m⁵C) modifications by slot blot assay

HEK293T and 2B2 cells were treated with doxycycline (1 µg/µL) for 3 days and reverse transfected with siNC, siNSUN2, or siDNMT2 (60 nM) and Lipofectamine RNAiMax. On the third day, cells were subjected to DNA damage via 10 Gy irradiation and incubated for 10 min at 37 °C. Nuclear RNA was subsequently isolated by cellular fractionation. Briefly, cells were resuspended in Hypotonic Buffer (20 mM Tris-HCl pH 7.4, 10 mM KCl, 3 mM MgCl₂, 0.1% NP-40 in DEPC-treated water) and incubated on ice for 5 min. Nuclei were pelleted by centrifugation at 500 × g for 5 min at 4 °C and lysed in Cell Lysis Buffer (50 mM Tris-HCl pH 7.4, 150 mM NaCl, 3 mM EDTA, 3 mM EGTA, 1% Triton X-100, 0.1% SDS, 2 mM DTT in DEPC-treated water) for an additional 5 min on ice. Lysates were mixed with TRIzol LS reagent (Ambion # 10296028) and 1-bromo-3-chloropropane (3:1, v/v), then incubated at 55 °C for 5 min using a thermomixer. RNA was isolated by isopropanol precipitation, washed with cold 100% ethanol, air-dried, and resuspended in TE buffer (low EDTA). Genomic DNA contamination was removed using Turbo DNase (Invitrogen, Cat. #AM2238) for 1 hour at 37 °C. For slot blot analysis, 500 ng of RNA was diluted in TE buffer (low EDTA) and applied to a pre-equilibrated (with TBS), positively charged nylon membrane (Amersham Hybond-N+, Cat. #RPN203B) using a slot blot apparatus (Bio-Rad, Cat. #1706542) under gentle vacuum. The membrane was washed once with TBS buffer, UV-crosslinked (0.2 mJ/cm²). and blocked with Protein-Free TBS Blocking Buffer (Pierce, Cat. #37570) for 1 h at room temperature. It was then

incubated overnight at 4 °C with anti-m⁵C primary antibody (1:500) in Protein-Free TBS Blocking Buffer, followed by a 1-h incubation with secondary antibody (1:2500) in Protein-Free TBS Blocking Buffer at room temperature. Signal detection was performed via auto-radiography. RNA membrane loading was verified using methylene blue staining. A complete list of antibodies used in this study is provided in Supplementary Data 1.

### R-loop detection in HEK293T and 2B2 cells

HEK293T and 2B2 cells were treated with Doxycycline (1 µg/mL) on Day 0 to induce shDicer expression and maintained under treatment for three days. On Day 1, cells were reverse transfected with 60 nM of either negative control siRNA (siNC), siRNA targeting NSUN2 (siN-SUN2), or siRNA targeting DNMT2 (siDNMT2) using Lipofectamine RNAiMAX, according to the manufacturer's instructions. On Day 2, cells were re-plated in 10 cm dishes to achieve ~60% confluency by Day 3. On Day 3, cells were exposed to 5 Gy of ionizing radiation and harvested 5 min post-irradiation. Genomic R-loops were isolated following the protocol described by Sarker et al.[54], with minor modifications. Briefly, prior to membrane transfer, R-loop-containing nucleic acids were treated overnight at 37 °C with RNase T1 and ShortCut RNase III (1:200 dilution). 500 ng of genomic R-loops were then transferred to a positively charged nylon membrane and crosslinked using UV irradiation (0.2 mJ/cm²). The membrane was blocked for 1 h at room temperature using Protein-Free TBS Blocking Buffer, followed by overnight incubation at 4 °C with the S9.6 antibody (1:1000) diluted in the same blocking buffer. The membrane was then incubated for 1 h at room temperature with the secondary antibody (1:2500) diluted in Protein-Free TBS Blocking Buffer. R-loop loading on the membrane was assessed using methylene blue staining. A complete list of antibodies used in this study is provided in Supplementary Data 1.

### Immunofluorescence

U2OS cells were reverse transfected with siNC and siNSUN2 (60 nM). After 24 h cells were transfected with FLAG-NSUN2 and FLAG-NSUN2-K190M. Then cells were seeded on poly-L-lysine coated coverslips to reach 60% confluency at the time of ionising irradiation. Cells were damage with 5 Gy and fixed (4% PFA in PBS for 10 min at 37 °C) at certain time points. After permeabilization (0.2% Triton X-100 in PBS for 10 min at room temperature) and blocking (5% BSA in PBS for 1 h at 37 °C) steps, FLAG and Phospho-Histone H2Ax (Ser139) primary antibodies (1:1000) were incubated in 5% BSA solution overnight at 4 °C. Then, coverslips were incubated with Alexa Fluor 488 (Invitrogen #A-11001) and Alexa Fluor 555 (Invitrogen #A-21428) secondary antibodies (1:2000) for 2 h in dark, sealed with transparent nail varnish and visualized with Olympus FV1200 confocal microscope with 60X Oil Immersion Objective. Foci were quantified using Cell profiler. A complete list of antibodies used in this study is provided in Supplementary Data 1.

### Laser micro-irradiation

HEK293T and HEK293T 2B2 were treated with Doxycycline (1 µg/µl) for 3 days and transiently transfected with either NeonGreen-NSUN2 or NeonGreen-NSUN2-K190M or NeonGreen-only. U2OS cells were transiently transfected with either NeonGreen-NSUN2 or NeonGreen-only. Then, cells were seeded onto 35 mm glass bottom dishes (Greiner #627860) to reach the confluency of 60 % at the time of imaging. Before laser-induced damage, cells were sensitized by incubation (30 min at 37 °C) with 10 µM Hoescht 33342 (Thermo Scientific #H3570). During laser micro-irradiation experiment cells were maintained in an incubation chamber at 37 °C and 5% CO₂. Laser stripes were generated on a Nikon Spinning Disk SoRa microscope by a 405 pulsed laser in loops of 80 iterations and average peak powers of 1.5 mW and 2.4 mW. Recruitment and retention of NSUN2 at the site of induced damage was then monitored by tracking the fluorescence

intensity at 488 nm (1 frame every 4 seconds). Time-elapse images were analysed using Fiji software.

### ChromRNASeq data processing

Adapters from ChromRNASeq libraries were trimmed using Cutadapt (version 4.4) (https://cutadapt.readthedocs.io/en/stable/installation.html) in paired-end mode. The quality of the resulting FASTQ files was assessed using FastQC (https://www.bioinformatics.babraham.ac.uk/projects/fastqc/). Trimmed reads were aligned to the human hg19 reference genome using STAR aligner. BAM files were then strand-separated into positive and negative alignments using Samtools (https://www.htslib.org/).

### Metagene plots

Strand-specific coverage files representing CPM-normalized read counts per nucleotide were generated using the *bamCoverage* function from deepTools (https://deeptools.readthedocs.io/en/develop/). The *computeMatrix* function was then used on strand-separated bigWig files to compute CPM coverage in the ±2.5 kb flanking regions around AsiSI sites, with a bin size of 1 nucleotide. *bedtools intersect* was used to identify gene regions overlapping these flanks. A custom Python script annotated each bin in the positive strand matrix as either sense or antisense, based on the orientation of nearby genes. The same logic was applied to the negative strand matrix. Only bins overlapping genes were used for annotation.

Sense bins from both positive and negative strand matrices were concatenated to form a unified sense matrix. Antisense matrices were constructed similarly. For each AsiSI site, read counts from overlapping genes were summed so that each row represented one site and each column one bin (total of 5000 bins for ±2.5 kb). Matrices were categorized based on site annotations provided by Prof. Legube (e.g., HR-prone, NHEJ-prone, uncut, transcriptionally active or inactive). Line plots and fill plots were generated using the *matplotlib* Python package.

### PCA plots

Read coverage within 5 kb of 80 AsiSI sites (BLESS-defined, courtesy of Prof. Legube) was summarized using *bigwigSummary* from deepTools and compared across replicates using the *plotPCA* function.

**Box plots**. CPM coverage values from sense and antisense matrices were summed across 500 bp flanking regions of each AsiSI site and visualized as box plots using *matplotlib*. Significance was tested using the two-sample Wilcoxon test from *scipy*. HR versus NHEJ coverage was compared using the Mann–Whitney test.

Log₂ fold changes were calculated as the ratio of CPM coverage in condition vs. control (for sense and antisense matrices separately). These were visualized as box plots and statistically assessed using the two-sided Wilcoxon test.

**Heatmaps**. The *plotHeatmap* function from deepTools was used to generate heatmaps centred on all annotated DSBs sorted by cleavage efficiency. Heatmaps were created separately for sense and antisense matrices.

**Cumulative distribution plots**. Cumulative distribution plots showing fold changes in read coverage ( ± 500 bp around AsiSI sites) between condition and control were generated using a custom Python script. Statistical significance between sense and antisense distributions was evaluated using the two-sample Wilcoxon test from *scipy*.

**BLESS-Seq data processing**. BLESS-Seq data (E-MTAB-5817) were processed as previously described. Read coverage within ±500 bp of DSBs was computed using *bedtools multicov*, and sites were ranked based on read count to estimate cleavage efficiency.

**ChIP-Seq data processing.** ChIP-Seq libraries were sequenced using Illumina NovaSeq X Plus 500 (paired-end, 150 bp). FastQC was used to assess read quality before and after trimming. Reads were aligned to the hg19 genome using BWA. Duplicate reads were removed, and alignments were sorted and indexed using Samtools (https://www.htslib.org/). *bamCompare* from deepTools was used to calculate $\log_2$ fold change (+4OHT vs. −4OHT). Peaks were called using MACS2 (https://github.com/macs3-project/MACS) with a $q$ value cutoff of 0.01 and required to be present in all three replicates.

**ChIP-Seq PCA plot.** Coverage around ±2.5 kb of AsiSI sites was quantified using *bedtools multicov*, normalized to CPM, and analyzed using the *PCA* function from *scikit-learn*.

**ChIP-Seq metagene plot.** The *computeMatrix* function from deep-Tools was used to generate average profiles of $\log_2$ fold changes (+4OHT/−4OHT) in ±2.5 kb regions around DSBs. Bin size was set to 1.

**ChIP-seq box plot.** Read coverage in ±2.5 kb regions of DSBs were quantified with *bedtools multicov* and normalized to CPM. Box plots were created in *matplotlib*, and the two-sided Wilcoxon test was used to determine significance (P < 0.01) using *scipy*.

**ChIP-seq IGV profiles and heatmaps.** CPM-normalized bigWig files were generated using *bamCoverage* and visualized in IGV. Heatmaps of all annotated DSBs were generated using *plotHeatmap* from deepTools.

**ONT Data Processing.** Oxford Nanopore RNA004 reads were base-called using Dorado (https://github.com/nanoporetech/dorado) with HAC and RNA modification models for Ψ, $m^5C$, $m^6A$, and inosine. Reads were aligned to the hg38 genome using the Dorado aligner with parameters "-ax splice -k 14". Modified bases were identified using *modkit pileup* (https://github.com/nanoporetech/modkit) with a modification threshold of 0.7. Output bedMethyl tables were used for all downstream analysis.

**ONT box plots.** Differential $m^5C$ modification between conditions was assessed using *modkit dmr pair*. Regions of interest included ±2.5 kb flanks of AsiSI sites and NSUN2 ChIP-Seq peaks. Counts of modified residues were visualized as box plots in *matplotlib* and statistically tested using the two-sided Wilcoxon test.

**ONT $m^5C$ metagene plot.** The *modkit localize* command was used to extract the percentage of $m^5C$ -modified residues at each position within ±2 kb around AsiSI sites and NSUN2 ChIP-Seq peak summits. Results were plotted using *matplotlib*.

**ONT snapshots.** Aligned ONT reads were visualized using IGV. Reads were grouped by strand, and $m^5C$ modification probabilities were shown using the "show modification" option.

**HR and NHEJ reporter assay**
HeLa cells stably expressing EJ5-GFP and DR-GFP cassette were reversed transfected with 60 nM of siNC, siNSUN2, siRAD51. After 24 h cells were seeded and forward transfected with pCBASceI plasmid (Addgene #26477)[55] and Lipofectamine 3000. After 48 h, cells were trypsinised and resuspended with 10%FBS in PBS before running FACS. siRAD51 was used as positive control for HeLa HR reporter cells; the DNA-PK inhibitor Wortmannin (Sigma, W3144-250UL, 1 μM for 6 h) was used as the positive control for HeLa NHEJ reporter cells. Samples were acquired by CytoFLEX Flow cytometer (Beckman Coulter) and analysed using FlowJo software.

**Comet assay**
5000 Hela cells previously transfected with 60 nM of siNC, siNSUN2 and siRAD51 were embedded in 0.5% CometAssay LMAgarose (Biotechne #4250-050-02) before being spotted on Cometslides (Biotechne #4250-050-03). On-gel cell lysis was performed by placing the Cometslide into lysis buffer (2.5 M NaCl, 0.1 M EDTA, 10 mM Tris-Base, 10% DMSO, 1% TritonX-100, pH=10) overnight at 4 °C. After wash with $ddH_2O$, Cometslides were immersed into running buffer (0.3 M NaOH, 1 mM EDTA, pH=13) for 1 h at 4 °C followed by gel electrophoresis at 300 mA for 30 min. Then, Cometslides were washed twice with Neutralization Buffer [0.4 M Tris-base buffer (pH=7.5)] for 5 min at room temperature, incubated with 70% ethanol for 15 min and air dried. For chromatin tail visualization, incubation of 2 μg/mL DAPI (BD Biosciences #564907) in PBS for 5 min followed by 5 min $ddH_2O$ washing was performed. Images were acquired by EVOS M7000 microscope with 10X objectives. To perform the tail moment quantification, Fiji with OpenComet plugin was used. Statistical significance was determined by using unpaired Welch's correction.

**Western blot**
Approximatively $1 \times 10^7$ cells were lysed in 200 μl of RIPA buffer (Thermo Scientific #89901) supplemented with 1x Protease inhibitor (Roche # 11873580001) and incubated at 4 °C for 20 min. Soluble supernatant was collected after centrifugation of 5 min at 500 g (4 °C), protein concentration was determined by Bradford colorimetric method (BioRad #5000006). Total proteins (20 μg) of whole cell extracts were resuspended in 1x Laemmli Buffer (BioRad #1610747), boiled for 10 min at 95 °C and examined by western blot using 4–15% Mini-PROTEAN TGX precast protein gels (BioRad #4561083). After blocking step with 10% non-fat milk for 1 h at room temperature, primary antibodies (1:1000) were incubated in 10% non-fat milk overnight at 4 °C. Secondary antibodies (1:20000) were incubated for 1 h at room temperature and blots were imaged with ECL substrate (Thermo Scientific #32106) and Amershan Hyperfilm ECL (Cytiva # 28-9068-35). A complete list of antibodies used in this study is provided in Supplementary Data 1.

**Cell cycle analysis**
Cells (after siRNA treatment) were treated with 10 μM EdU (5-ethynyl-2'-deoxyuridine).

(Invitrogen, A10044) for 30 min before trypsinization and fixed with 70% ethanol overnight. After fixation, cells were permeabilised in PBS with 0.1% Triton X-100 (Sigma, X100-500ML) and 1% BSA (Sigma, 10735086001) for 15 min at 4°C; followed by further treatment with PBST-BSA buffer [PBS containing 0.1% Tween-20 (Sigma, P7949-500mL) and 1% BSA] for 5 min at 4°C twice. The Click-IT reaction [2 mM CuSO4 (Sigma, C8027-500g), 10 mM Sodium Ascorbate (Sigma, A7631-25g), 10 μM Alexa fluor 555 (Invitrogen, A20012) in PBS (pH=7.2)] was performed in dark for 1 h at 4 °C. After wash cells with PBST-BSA buffer twice, cells were counter-stained with 1 μg/ml DAPI (BD Biosciences, 564907) in PBS with 0.1% BSA, 0.1 mg/ml RNase A (Thermo scientific, EN0531) for 1 h in dark at room temperature. Cell cycle distribution was analysed using a CytoFLEX LX (Becton Dickinson) equipped with the CytExpert programme and the obtained datasets analysed by FlowJo software.

**MTS proliferation assay**
For cellular viability analysis, 96-well plates were seeded with 5,000 HeLa cells previously transfected with siNC, siRAD51 and NSUN2 and damaged with 5 Gy. The MTS assay was carried out at certain time points using The CellTiter 96 AQueous One Solution Cell Proliferation Assay (Promega, #G3580) according to the manufacturer's instructions, and absorbance was measured at 490 nm.

## Statistical analysis

Statistical analyses were performed in GraphPad Prism. The Kolmogorov-Smirnov normality test was performed to test for a normal distribution. If the data met the criteria for a normal distribution, statistical analysis was performed using two-way ANOVA or unpaired t-test with Welch's correction. For data that did not follow a normal distribution, non-parametric tests including the two-tailed Mann−Whitney U test or the two-sided Wilcoxon test were applied. Immunofluorescence assays were conducted in three independent biological replicates ($n = 3$), with a total of at least 100 nuclei evaluated per condition. Immunofluorescence, PLA and FISH-PLA data are presented as mean values ± standard deviation (SD), statistical significance was assessed using two-tailed, non-parametric Mann−Whitney test, unless otherwise specified. Exact $P$ values are reported in the corresponding figure legends. For all box plots presented in the manuscript, statistical comparisons of medians between sense and antisense $\log_2$ fold change distributions were performed using a two-sided Wilcoxon signed-rank test. Box plots display the median (central line), the interquartile range (25th to 75th percentiles, box), and whiskers represent the minimum and maximum values. Exact p-values are shown in the figures. All analyses were conducted with n = 3 biological replicates. Exact $P$ values are reported in the corresponding figure legends, unless otherwise indicated.

## Reporting summary

Further information on research design is available in the Nature Portfolio Reporting Summary linked to this article.

## Data availability

ChrRNA-seq data have been deposited to NCBI's Gene expression Omnibus and are accessible under GEO Series accession number GSE260748. NSUN2 ChIP-seq data have been deposited to NCBI's Gene expression Omnibus and are accessible under GEO Series accession number GSE294470. Oxford Nanopore data have been deposited to NCBI's Gene expression Omnibus and are accessible under GEO Series accession number GSE304944 (https://www.ncbi.nlm.nih.gov/geo/query/acc.cgi?acc= GSE304944). The data reported and any additional information required to re-analyse the data reported in this study is available from the corresponding author upon request. Source data are provided with this paper.

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

## Acknowledgements

We thank all members of Gullerova lab for their support. We are grateful to Dr Esra Balikci and Prof Kilian Huber for their help with preliminary experiments and Oxford Nanopore Technologies for their ongoing support and advice. This work was supported by the Senior Research Fellowship by Cancer Research UK [grant number BVR01170], Medical Research Council [BVR02590], EPA Trust Fund [BVR01670], and Lee Placito Fund awarded to M.G.

## Author contributions

A.A. designed and performed most of the experiments and prepared initial draft of the manuscript. A.D.F. purified NSUN2 and DICER proteins. K.A. performed the bioinformatic analysis of Chr-RNA-seq, ChIP-seq and Nanopore sequencing. Q.L. performed comet, MTT and reporter assays. M.G. designed and supervised the project and wrote the manuscript. All the authors reviewed and approved the final version of the manuscript.

## Competing interests
The authors declare no competing interests.
