## [Transparent Peer Review file · Nature Communications]

NSUN2 Facilitates DICER Cleavage of DNA Damage-Associated R-Loops to Promote Repair

Corresponding Author: Professor Monika Gullerova

Version 0:

Reviewer comments:

Reviewer #1

(Remarks to the Author)

In this work, Alagia et al. describes the involvement of the m5C-methyltransferase NSUN2 and RNA methylation in DNA repair. Mechanistically, NSUN2 is recruited to DNA damage sites to methylate DARTs, which facilitates removal of DART-associated R-loops by DICER. The paper is well-written, the results novel and interesting and the manuscript appropriate for the broad readership of Nature Communications. But before publication, the authors need to strengthened their results by including additional controls for the PLA technique and establishing the effects of NSUN2 depletion on formation and fate of DARTs.

These are my major concerns:

Figure 1

Accumulation of NSUN2 at DNA damage sites is convincing, but as the rest of the study focuses on the DS2 site, it would be good to examine enrichment of NSUN2 at this site by ChIP instead of current DS1 site.

Figure 2

I feel formation of DARTs, methylation of these transcripts and interaction between NSUN2 and DARTs are not demonstrated convincingly. First, the authors need to ensure that the PLA probe (DS2 AS) used for detection of DART is not hybridizing to DNA at break sites. The enhanced signal after DNA damage could reflect increased probe binding to end-resected DNA and the effects of TLP might be due to reduced resection. The authors need to show that the PLA signal between DS2 AS and NSUN2 (or DICER) is removed by RNase A or RNase H treatments. This could also give insight into the proportion of DARTs involved in hybrid formation.

Secondly, the authors need to show that DARTs are transcribed from break sites by performing co-staining of the PLA (DS2 AS + NSUN2) with a marker of DNA breaks, e.g., γH2AX or 53BP1.

The author claims that incorporation of 5-azacytidine (zC) results in trapping of NSUN2 protein on DARTs but not data to support this is shown. This needs to be backed up experimentally, for example by examining retention time of NSUN2 at laser stripes in cells pre-treated with zC and/or performing PLA between NSUN2 and DART in the absence of zC.

Interaction between NSUN2 and DART needs to be strengthened by another method, for example immunoprecipitation of NSUN2 (CLIP or RIP) in DiVA cells followed by qPCR for DART. This could be done in combination with zC, to support the claim that NSUN2-DART interaction is enhanced by zC.

The authors say that NSUN2 methylates DARTs but this has not been shown. The authors could perform a PLA between 5mC antibody and the DART probe to visualize accumulation of 5mC at break sites. Alternatively combine 5mC with a DNA repair factor in PLA. Thereafter verify that the PLA signals from 5mC are reduced by siNSUN2.

Figure 3

Looks good

Figure 4

Interaction between NSUN2 and DICER should be strengthened using immunoprecipitation (IP). This can be done using recombinant proteins in vitro or between endogenous proteins in cells. In vitro, it would be interesting to see whether addition of RNA enhances the interaction.

Are the PLA signals between NSUN2 and DICER after IR occurring at sites of DNA damage and are they reduced by siNSUN2 or siDICER?

Can interactions between NSUN2 and DICER be detected after 4OHT in the DiVA AsiSI cells?

To make sure RNases do not impair PLA signals indirectly, a PLA experiment between other repair factors whose interaction does not depend on RNases should be included as control.

Figure 5

Is the PLA signal for DS2 AS and DICER reduced by RNase A or RNase H treatments?

Can interaction between DICER and 5mC be detected by PLA?

Can accumulation of DICER at DNA damage sites be shown by another method (laser stripes or ChIP)? Alternatively, PLA with gH2AX. And is accumulation of DICER at DNA damage sites altered following pre-treatment with zC?

Figure 6

R-loop substrates generated in vitro should be verified, or at least subjected to RNase A treatment to remove unbound RNA before the DICER cleavage reaction. Otherwise, cleavage of unbound RNA (not part of R-loop) by DICER might contribute to the results in 6A-B.

It has not been tested whether NSUN2 methylates DS2 AS RNA in vitro and the mechanism by which NSUN2 facilitates removal of DS2 AS-associated R-loops by DICER remains unclear. To provide insight here, it would be interesting to examine whether pre-methylated R-loops is a more efficient substrate for DICER-mediated cleavage. Pre-methylated DS2 AS RNA can be generated by including m5C nucleotides instead of C during the IVT reaction, followed by formation of R-loop substrates.

6C: RNase H should be included as a negative control in the PLA between NSUN2 and S9.6.

As primary DARTs are increased by siNSUN2, it would be important to examine whether R-loops and associated DICER are elevated at break sites following siNSUN2 (PLA: S9.6 + DICER). Conversely, whether overexpression of NSUN2, either transiently or stably (in the HEK293 cells used in Fig 1B) results in reduced formation of R-loops.

Figure 7

Looks good.

Reviewer #2

(Remarks to the Author)

In the present study, the authors investigated the role of NSUN2, an RNA m5C-methyltransferase, in DSB repair at transcriptionally active loci. The authors attempted to demonstrate the involvement of NSUN2 in DICER-associated RNA processing during DSB repair. The authors propose that NSUN2 is involved in the DSB repair of transcriptionally active loci; however, several important control experiments were not performed in this study. The authors should address all of the concerns below.

1. A PLA assay using NSUN2-gH2AX antibodies was conducted to show the recruitment of NSUN2 to DSB sites. However, H2AX phosphorylation occurs only after DNA damage, i.e., H2AX is not phosphorylated in non-irradiated cells. Therefore, the absence of the PLA (NSUN2-gH2AX) signal in non-irradiated cells is to be expected. And, the PLA signal may have increased owing to non-specific signals because a large amount of H2AX is phosphorylated after high-dose IR. How can the authors verify that the PLA signal is specific to the NSUN2-gH2AX interaction? A major problem is that the authors used a dose of 10 Gy in this assay. A dose of 10 Gy induces 300 DSBs per cell in the G1 phase and 600 DSBs in the S/G2 phase. Despite 300-600 DSBs, only 10 PLA signals (15 (after IR) – 5 (background)) were induced, i.e., the NSUN2-gH2AX interaction occurred only at <3% of the DSB sites. The authors should address these concerns. In addition, the authors should present a representative image of gH2AX IF and the overlap of gH2AX foci with the PLA signal.

2. The laser track quality is poor. On a laser track similar to this image, many SSBs and gaps, which are repaired by BER or NER, are generated. The author should show that the laser condition primarily induces DSBs, or that it contains SSBs and gaps as well as DSBs. In addition, the gH2AX or 53BP1 signal obtained by IF or live imaging should be shown at the laser track. GFP alone should also be shown because such high-power lasers can recruit GFP fragments.

3. Why did the authors use TPL in Fig. 1 and DRB in Fig. 4? In general, because transcription inhibitors block transcription with different mechanisms, multiple transcription inhibitors are tested. Both TPL and DRB should be tested, at least in Fig 1 and Fig. 4

4. In Fig. 1b, why did the authors not examine the left side of the AsiSI site? Transcription inhibitors should also be tested.

5. The authors preferred to use a PLA assay to demonstrate the interaction in this study. PLA is a tool used to detect protein-protein (or nucleotides) interactions under a microscope; however, non-specific signals are often detected by PLA. The authors should confirm these interactions using other methods such as a co-IP. In addition, for example, NGS analysis in combination with co-IP can also be applied by using the AsiSI system.
6. The authors have mentioned that BLESS was performed in the manuscript (Fig. 3a). A heatmap of BLESS should be included, because siNSUN2 may change the distribution of DSB induction. In addition, the authors should present the results of the sense/antisense reads of the 4-OHT sample. In the current version, the scale of the x-axis is +/- 2.5 kbp. The authors should show a wider area because the RNA signal may spread over 2.5 kbp from the AsiSI site in siNSUN2 cells.
7. Fig. 3. f–g: the result for the low transcription DSB site should be shown.
8. In Figs 4–6, whether this is a DSB-dependent interaction is unclear. Only a few PLA foci were induced after irradiation with 10 Gy or approximately 300 DSBs. This interaction should be confirmed using a PLA-independent method.
9. In Fig. 7, the authors performed the comet, cell survival, and DR-GFP assays in RAD51- or NSUN2-depleted cells. The magnitude of repair defect in siNSUN2 cells is very minor although it is statistically significant. In particular, the defect in the DR-GFP assay is very small. Therefore, whether these results provide evidence that NSUN2 promotes HR is unclear. To further confirm this data, RAD51 foci per cell should be examined. In addition, double knockdown of siNSUN2+siRAD51 should be performed to address whether it is epistatic. The cell cycle profile should be shown because a low percentage of the S/G2 phase affects HR efficiency in the DR-GFP assay. Why did the authors use BRCA1 siRNA in DR-GFP? The siRAD51 should be used.
10. Time course should be performed to examine DSB repair defect using gH2AX foci analysis. The initial induction of gH2AX foci formation may be changed in the NSUN2 mutant.
11. The use of the word of “DART” by the authors is confusing because the concept of DNA damage-induced RNA is reported by Fabrizio d’Adda di Fagagna as damage-induced long non-coding RNAs (dilncRNAs).

Reviewer #3

(Remarks to the Author)

The study investigates the molecular mechanisms of NSUN2, an RNA m5C-methyltransferase involved in the repair of double-stranded breaks (DSB). It was found that NSUN2 localizes to DSBs in a transcription-dependent manner, and binds to and methylates DARTs, which are DNA damage response RNAs. Furthermore, they detected an RNA-dependent interaction between NSUN2 and DICER, which was stimulated by DNA damage. NSUN2 activity was shown to promote DICER cleavage of DARTs-associated R-loops, essential for efficient DNA repair. The topics of the study are interesting. However, I found that the study did not convincingly demonstrate the working models of NSUN2. Below are my specific comments.

1. In Figure 1a, the Proximity Ligation Assay (PLA) included a single antibody control for NSUN2 under -IR conditions and for γ H2AX under +IR conditions. Ideally, they should have included single antibody controls for both proteins under both -IR and +IR conditions to ensure comprehensive control data. Additionally, the plot lacks clear indications of what the y-axis represents, and it is not specified what each dot in the plot signifies.
2. In Figure 1b, the authors introduced TPL into the experiment without providing an explanation for its inclusion. Further clarification is needed on why TPL was used in this context. Additionally, the correctness of the "Relative Fluorescence Signal" equation mentioned in the figure legend should be confirmed for accuracy.
3. Moreover, the authors need to describe in more detail the differences in the fluorescence signal in the cell images before and after damage. Specifically, an explanation of how the fluorescence signal disappears upon laser treatment would aid in understanding the dynamics involved. Furthermore, to substantiate the role of TPL in this experiment, it is essential to include an examination that confirms whether TPL treatment results in transcription inhibition effectively.
4. In Figure 1c, the authors examined the localization of NSUN2 at the DSB using the DiVA system. However, they only investigated the presence of NSUN2 at the DS1 locus. To provide a more comprehensive analysis, it would be beneficial for the authors to conduct sequencing to examine the localization of NSUN2 on a genome-wide scale and verify this localization across several loci.
5. In Figure 2, the authors utilized zC-FISH-PLA to examine the interaction of NSUN2 with DARTs at two loci, DS2 and DS3. The results indicated that the NSUN2 protein interacted with antisense DARTs at DS2 but not with DS3 itself. It is unclear why the authors chose to conduct these assays using DS2 and DS3 instead of DS1. Additionally, NSUN2 was only found at DS3, prompting concerns about whether the zC-FISH-PLA technique is sensitive enough to detect interactions between NSUN2 and DARTs, or if NSUN2 can only interact with certain DARTs. Given that the interaction was confirmed at only one locus (DS2), it is premature to conclude generally that NSUN2 binds to antisense DARTs based on this single instance. Further investigation across more loci and with enhanced sensitivity in detection methods would help in establishing a more robust conclusion.
6. The authors stated that the formation of the zC-RNA-NSUN2 complex signifies the active addition of m5C to RNA transcripts. To substantiate this claim, they should provide additional details on how this process occurs.
7. In Figure 2b, it is necessary to include a Western blot analysis to verify the NSUN2 protein levels after knockdown (KD).
8. It is essential to provide evidence confirming that NSUN2 has been effectively knocked down and that 4-hydroxytamoxifen (4-OHT) is functional. This could include data from Western blot analyses to demonstrate the reduction in NSUN2 protein levels post-knockdown, and functional assays or reporter gene activity to verify the efficacy of 4-OHT.

9. Why was there a sudden switch to using DRB instead of continuing with TPL? It would be helpful for the authors to explain the rationale behind this change in the choice of inhibitors.

10. In Figures 6a and b, where it is claimed that NSUN2 facilitates DICER processing of DART-associated R-loops, this conclusion should be carefully reevaluated. Firstly, to ensure the purity of the used DICER enzyme and to exclude any nuclease contamination, it is recommended that the authors conduct cleavage assays using some pre-miRNAs as quality control experiments. Secondly, the methodology involving qPCR to examine the original RNA and the remaining RNA after cleavage using three different pairs of primers appears to yield inconsistent results. Therefore, the interpretation of these results should be reconsidered. The authors could also run the RNA on a gel to directly determine the cleavage efficiency of DICER, which would provide a more direct method for quantification. These additional experiments and clarifications would strengthen the validity of the claim regarding NSUN2's role in facilitating DICER processing of R-loops.

11. In Figure 6d, it is necessary to include at least three experimental repeats to ensure the robustness and reproducibility of the results.

Version 1:

Reviewer comments:

Reviewer #1

(Remarks to the Author)

The authors have done a good job in addressing my previous concerns. These new data now strengthen the claims and conclusions of the manuscript, and I recommend publication of this study in Nature Communications.

Reviewer #2

(Remarks to the Author)

The authors did not properly understand my original request regarding the γ H2AX immunofluorescence. As I clearly stated in my previous review, 10 Gy induces approximately 300–600 DSBs per nucleus. At 15 minutes post-irradiation, the majority of these breaks are still unrepaired. Consequently, staining with γ H2AX at this time point results in pan-nuclear signal distribution, meaning that virtually any nuclear protein may appear to colocalize with γ H2AX simply due to signal saturation. To illustrate this point, the authors should provide a representative γ H2AX IF image at 15 minutes post-10 Gy. Furthermore, because Figure 1a shows 20–60 PLA signals per nucleus, a dose-response experiment (e.g., 2, 5, 10 Gy=10, 30, 60 PLA signal should be detected) should be performed to demonstrate specificity and dynamic range. Importantly, the colocalization of γ H2AX foci with PLA signals must be shown using samples post-IR, not AsiSI-induced breaks.

Again, the authors did not address my request adequately regarding the laser track system. As I previously pointed out, the laser irradiation induces not only DSBs but also SSBs and base damage. Therefore, the authors must validate whether this laser system predominantly induces DSBs, rather than other types of damage. This could be achieved by testing the recruitment of SSB repair or BER factors. Although NSUN2 recruitment to the AsiSI site was shown, the CPM increase is very modest. If NSUN2 is mainly recruited to SSBs or base damage rather than DSBs in the laser track system, the conclusions drawn from these experiments could be misleading.

In Figure 3, there is no textual description of the results involving Exonuclease VII or RNase H in the Results section. More importantly, the observed induction of the PLA signal is marginal, making it difficult to evaluate whether RNase H treatment significantly attenuates the PLA signal. In fact, the RNase H-treated sample post-4OHT still shows approximately 3–5 PLA foci, raising serious concerns about the specificity of the observed signal reduction.

Regarding Figures 3d and 5e, it is unclear whether the presented images are projections or single z-slices. The authors should provide representative monolayer images or 3D reconstructions to verify spatial relationships between PLA and γ H2AX foci.

The magnitude of the DNA repair defects observed in the HR and NHEJ assays, as well as in the γ H2AX assay, appears to be marginal. It remains unclear whether these small differences have biological significance. The number of γ H2AX foci should be quantified, rather than relying solely on intensity measurements. In addition, although the 4 h and 24 h images in the control condition are relatively dark, visible foci remain, suggesting the authors may have overinterpreted the results.

Lastly, for Extended Figure 2h, box plots for additional transcriptionally active and inactive sites should be provided to support the generality of the findings.

Reviewer #3

(Remarks to the Author)

I have no further comments. The responses from the authors are good. Thank you.

Version 2:

Reviewer comments:

Reviewer #2

(Remarks to the Author)

I acknowledge the authors' responses to my concerns. I do not have any further comments.

Reviewer #4

(Remarks to the Author)

As requested, I have focused in the exchange between reviewer 2 and the authors.

Regarding comments from reviewer 2:

1. About gH2AX foci and colocalization.

I agree with the reviewer that the number of breaks at that dose and timepoint result in a pan nuclear staining, so talking about co-localization does not makes sense. Everything in the nucleus will "colocalize" as the whole nucleus will be covered with gH2A signal. The suggestion from the authors using PLA has the same problems. PLA is a proximity assay, so everything will be close to a phosphorylated H2AX on that scenario. The use of HOXD11 as a negative control is not enough, as there might be other issues at play that can avoid the PLA signal. The dose response is the most informative experiment. On its own, the recruitment by foci is not fully conclusive, but overall, and considering all the different experiments together (see below), they have a strong case for their statement.

2. Representative image at 15 min.

The authors provided the dose response.

3. Laser tracks. I agree with the authors. It is true that laser microirradiation cause base damage, SSBs and DSBs. But the proposed experiment will only reflect that. I am convinced that all types of repair proteins will be recruited. But this will not mean that NSUN2 is bound to one or the other, as all are there. Considering all the other experiments, as the authors state, is what can prove that NSUN2 is bound to DSBs. The authors, however, can include a sentence stating that they cannot exclude that this protein might also bind SSBs or other types of DNA damage.

4. About figure 3: The authors have provided the context in the main text.

5. Regarding Figures 3d and 5e: The information is provided as requested.

6. Regarding the reporter systems: It is true that the systems have limitations, as stated by the authors, but the effect is extremely mild. I am very surprised that they have such a narrow error bar and can have this significance, as the reporters have also quite an intrinsic variability. The important thing here, in alignment with the authors model, is that NSUN2 probably is only engaged only a small subset of breaks (Transcriptionally active, with DARTs, etc), so the reporters are not the best way go. The comet gives support to the effect in DNA repair, but a control just after irradiation is missing. It is formally, although unlikely, possible that in the absence of NSUN2 radiation creates more breaks due to changes in the chromatin.

7. Quantification of gH2AX: Done by the authors.

8. Figure 2h: Done as requested

Overall, although there are individual experiments that have weaknesses, in the context of the whole manuscript it seems that the conclusions from the authors are supported by the experimental data. I understand the claims made by the referee, and in principle agree with many of them, but I have to side with the authors this time

Point-by-point response to the referees' comments.

Reviewer #1 (Remarks to the Author):

In this work, Alagia et al. describes the involvement of the m5C-methyltransferase NSUN2 and RNA methylation in DNA repair. Mechanistically, NSUN2 is recruited to DNA damage sites to methylate DARTs, which facilitates removal of DART-associated R-loops by DICER. **The paper is well-written, the results novel and interesting and the manuscript appropriate for the broad readership of Nature Communications.** But before publication, the authors need to strengthened their results by including additional controls for the PLA technique and establishing the effects of NSUN2 depletion on formation and fate of DARTs.

We are very grateful for this reviewer's helpful comments and appreciation of this interesting and novel manuscript. Below we describe how we addressed all their comments.

These are my major concerns:

Figure 1

Accumulation of NSUN2 at DNA damage sites is convincing, but as the rest of the study focuses on the DS2 site, it would be good to examine enrichment of NSUN2 at this site by CHIP instead of current DS1 site.

We appreciate this comment, and in order to fully address the presence of NSUN2 at DSBs genome wide (not only at selected DSBs), we have performed modified chromatin immunoprecipitation assay using NSUN2 antibody. In detail, we treated cells with 5-aza-C, which is an RNA analogue that specifically traps NSUN2 protein on the RNA substrate when subjected to methylation. This has been described previously (Hussain et al., Cell Reports, 2013).

Furthermore, we double crosslinked the cells using dithiobis (succinimidyl propionate (DSP) and formaldehyde (FA), both commonly used for crosslinking of protein to DNA, as suggested here: <https://www.abcam.com/en-us/technical-resources/protocols/dual-x-chip>.

Now we show NSUN2 distribution on chromatin genome wide at DSBs, see Figure 2a-e and Extended Data Figure 2.

Figure 2

I feel formation of DARTs, methylation of these transcripts and interaction between NSUN2 and DARTs are not demonstrated convincingly. First, the authors need to ensure that the PLA probe (DS2 AS) used for detection of DART is not hybridizing to DNA at break sites. The enhanced signal after DNA damage could reflect increased probe binding to end-resected DNA and the effects of TLP might be due to reduced resection. The authors need to show that the PLA signal between DS2 AS and NSUN2 (or DICER) is removed by RNase A or RNase H treatments. This could also give insight into the proportion of DARTs involved in hybrid formation.

As suggested, we have repeated FISH PLA for NSUN2/DARTs with additional controls. Specifically, we included treatment with Exonuclease VII to show that our probe is not hybridising with post resection overhang DNA. We also included treatments with RNaseH to show that NSUN2 is binding DARTs that are associated with R-loops. New data are now shown in new Figure 3b and c and Extended Data Figure 5a-c.

Secondly, the authors need to show that DARTs are transcribed from break sites by performing co-staining of the PLA (DS2 AS + NSUN2) with a marker of DNA breaks, e.g., gH2AX or 53BP1.

As suggested, we have performed FISH-PLA (NSUN2/DARTs) in combination with gH2AX IF. New data are shown in Figure 3d and for Dicer in Figure 5e and Extended Data Figure 11e.

The author claims that incorporation of 5-azacytidine (zC) results in trapping of NSUN2 protein on DARTs but not data to support this is shown. This needs to be backed up experimentally, for example by examining retention time of NSUN2 at laser stripes in cells pre-treated with zC and/or performing PLA between NSUN2 and DART in the absence of zC.

As suggested, we have performed NSUN2/DARTs FISH-PLA in the absence of zC, this is shown in Extended Data Figure 5d.

Furthermore, we have now performed zC NSUN2 ChIP-seq and also provide further explanation of zC mediated trapping of NSUN2 protein to RNA substrate in text and in Figure 2a. The experimental validation of zC trapping NSUN2 has been shown previously in (Hussain et al., Cell Reports, 2013).

Interaction between NSUN2 and DART needs to be strengthened by another method, for example immunoprecipitation of NSUN2 (CLIP or RIP) in DiVA cells followed by qPCR for DART. This could be done in combination with zC, to support the claim that NSUN2-DART interaction is enhanced by zC.

5-aza-Cytosine (5-aza-C) is a cytosine analogue that can trap RNA methyltransferases like NSUN2 by forming an irreversible covalent complex with the enzyme. NSUN2 normally catalyses the methylation of cytosine at the carbon-5 (C5) position in RNA to form 5-methylcytosine (m⁵C), a modification important for RNA stability, translation, and stress responses. This process involves the enzyme recognizing its RNA substrate, where a catalytic cysteine in NSUN2 performs a nucleophilic attack on the C6 position of the cytosine ring, forming a covalent intermediate. A methyl group from S-adenosylmethionine (SAM) is then transferred to the C5 position of cytosine, after which the enzyme is released. However, when 5-aza-C is incorporated into RNA in place of cytosine, its nitrogen atom at the C5 position prevents the methyl group from being transferred, blocking the completion of the methylation reaction. NSUN2 still forms the initial covalent intermediate with 5-aza-C at the C6 position, but because the methylation step cannot proceed, the enzyme becomes irreversibly trapped on the RNA. This mechanism is exploited experimentally to identify RNA targets of NSUN2 by "trapping" the enzyme at its binding sites. As DARTs are in close proximity to chromatin, this experimental approach further strengthens our conclusion that NSUN2 **binds and methylates** DARTS.

Furthermore, we include pioneering data showing Direct RNA sequencing using Oxford Nanopore Technology, showing NSUN2 dependent methylation on DARTs. Direct RNA sequencing using Oxford Nanopore Technology (ONT) enables the detection of RNA modifications like 5-methylcytosine (m⁵C) by analysing the characteristic changes they induce in the ionic current as RNA strands pass through the nanopore. Unlike traditional sequencing methods that require reverse transcription or amplification, ONT reads native RNA molecules directly, preserving modification information. Modified bases like m⁵C alter the electrical signal differently compared to unmodified cytosine, allowing trained machine learning models to identify and distinguish these modifications with increasing accuracy. Furthermore, we have validated these data in siNSUN2 samples.

Please see Figure 2 and Extended Figures 2, 3 and 4.

The authors say that NSUN2 methylates DARTs but this has not been shown. The authors could perform a PLA between 5mC antibody and the DART probe to visualize accumulation of 5mC at break sites. Alternatively combine 5mC with a DNA repair factor in PLA. Thereafter verify that the PLA signals from 5mC are reduced by siNSUN2.

We tried to use m⁵C antibody in PLA. Unfortunately, m⁵C antibody cannot distinguish DNA from RNA in PLA settings. We tried several approaches to obtain specific signal, but these didn't work. Therefore, instead we set up and optimised Oxford Nanopore technology to show directly NSUN2 dependent methylation of DARTs in Figure 2.

Figure 3

Looks good

Thank you :-)

Figure 4

Interaction between NSUN2 and DICER should be strengthened using immunoprecipitation (IP). This can be done using recombinant proteins in vitro or between endogenous proteins in cells. In vitro, it would be interesting to see whether addition of RNA enhances the interaction.

Indeed, we have purified both NSUN2 and Dicer recombinant proteins and performed quantitative Surface Plasmon Resonance assay to show that these two proteins interact with each other in concentration dependent manner. We prefer SPR approach, because it is

commonly used technique for detecting protein-protein interactions, in real-time and in natural state, providing more biologically relevant results. It offers detailed quantitative data, including binding strength (affinity), interaction rates (kinetics), and specificity. SPR is also highly sensitive, capable of detecting even weak or short-lived interactions that might be missed by other techniques.

These data are shown in Figure 5a.

Are the PLA signals between NSUN2 and DICER after IR occurring at sites of DNA damage and are they reduced by siNSUN2 or siDICER? Can interactions between NSUN2 and DICER be detected after 4OHT in the DiVA AsiSI cells?

To address this comment, we have performed PLA NSUN2/Dicer in AsiSI system and included siNSUN2 control. See new data in Figure 5b and Extended Data Fig. 10a.

To make sure RNases do not impair PLA signals indirectly, a PLA experiment between other repair factors whose interaction does not depend on RNases should be included as control.

As a control we have performed PLA using 53BP1 and gH2AX antibodies and treated samples with RNases. These data are shown in Extended Data Figure 10b.

Figure 5

Is the PLA signal for DS2 AS and DICER reduced by RNase A or RNase H treatments?

To address this comment, we have performed Dicer FISH-PLA and included RNaseH treatment to show that the Dicer interacts with DARTs associated with R-loops. These data are shown in Extended Data Figure 11b.

Can interaction between DICER and 5mC be detected by PLA?

Unfortunately, m5C antibody cannot distinguish DNA from RNA in PLA settings and is unsuitable for PLA.

Can accumulation of DICER at DNA damage sites be shown by another method (laser stripes or CHIP)? Alternatively, PLA with gH2AX. And is accumulation of DICER at DNA damage sites altered following pre-treatment with zC?

We have previously showed Dicer presence at DSBs by chromatin immunoprecipitation genome wide (Burger et al., JCB, 2017). We have now added these data to show the overlap between Dicer and DARTs, see new Figure 5d.

Figure 6

R-loop substrates generated in vitro should be verified, or at least subjected to RNase A treatment to remove unbound RNA before the DICER cleavage reaction. Otherwise, cleavage of unbound RNA (not part of R-loop) by DICER might contribute to the results in 6A-B.

As suggested, we have now repeated the preparation of the R-loops substrate and subjected it to RNaseH cleavage and S9.6 slot blot analysis. We now show that indeed our in vitro R-loop substrate is cleaved by RNaseH and recognised by S9.6 antibody therefore can be considered as a bona fide R-loop. These data are now added to Extended Data Figure 15c.

It has not been tested whether NSUN2 methylates DS2 AS RNA in vitro and the mechanism by which NSUN2 facilitates removal of DS2 AS-associated R-loops by DICER remains unclear. To provide insight here, it would be interesting to examine whether pre-methylated R-loops is a more efficient substrate for DICER-mediated cleavage. Pre-methylated DS2 AS RNA can be generated by including m5C nucleotides instead of C during the IVT reaction, followed by formation of R-loop substrates.

As suggested, we performed Dicer cleavage reaction with pre-methylated R-loop structures and indeed show increased activity of Dicer on pre-methylated R-loops, in comparison to un-methylated R-loops. These data are added to Figure 6a.

6C: RNase H should be included as a negative control in the PLA between NSUN2 and S9.6.

To address this comment, we have added RNaseH control to NSUN2/S9.6 PLA experiment, as shown in new Figure 6b.

As primary DARTs are increased by siNSUN2, it would be important to examine whether R-loops and associated DICER are elevated at break sites following siNSUN2 (PLA: S9.6 + DICER). Conversely, whether overexpression of NSUN2, either transiently or stably (in the HEK293 cells used in Fig 1B) results in reduced formation of R-loops.

We attempted to address this comment but unfortunately, Dicer antibody in combination with S9.6 antibody do not work for PLA (we only observed very few positive foci).

Instead, we now present slot blot using S9.6 antibody showing the levels of R-loops in absence of NSUN2 and/or Dicer upon DNA damage, please see Fig. 6c.

Figure 7

Looks good.

Thank you!

Reviewer #2 (Remarks to the Author):

In the present study, the authors investigated the role of NSUN2, an RNA m5C-methyltransferase, in DSB repair at transcriptionally active loci. The authors attempted to demonstrate the involvement of NSUN2 in DICER-associated RNA processing during DSB repair. The authors propose that NSUN2 is involved in the DSB repair of transcriptionally active loci; however, several important control experiments were not performed in this study. The authors should address all of the concerns below.

We are grateful for all these comments, which are all addressed in the revised manuscript.

We hope that this reviewer will find our revised manuscript interesting and suitable for Nature Communication readership.

1. A PLA assay using NSUN2-gH2AX antibodies was conducted to show the recruitment of NSUN2 to DSB sites. However, H2AX phosphorylation occurs only after DNA damage, i.e., H2AX is not phosphorylated in non-irradiated cells. Therefore, the absence of the PLA (NSUN2-gH2AX) signal in non-irradiated cells is to be expected. And, the PLA signal may have increased owing to non-specific signals because a large amount of H2AX is phosphorylated after high-dose IR. How can the authors verify that the PLA signal is specific to the NSUN2-gH2AX interaction? A major problem is that the authors used a dose of 10 Gy in this assay. A dose of 10 Gy induces 300 DSBs per cell in the G1 phase and 600 DSBs in the S/G2 phase. Despite 300-600 DSBs, only 10 PLA signals (15 (after IR) – 5 (background)) were induced, i.e, the NSUN2-gH2AX interaction occurred only at <3% of the DSB sites. The authors should address these concerns. In addition, the authors should present a representative image of gH2AX IF and the overlap of gH2AX foci with the PLA signal.

We appreciate this comment. First of all, PLA is an antibody-based assay and very much depends on quality and efficiency of an antibody to bind to its target. Additionally, there is a ligation step, generating ultimately the signal, which could be limited by the efficiency of the ligase. Furthermore, not all DSBs will be occupied by NSUN2 at one time. NSUN2, being an enzyme will interact with DSBs only transiently. Therefore, these limitations all together might limit the number of PLA foci observed. Although only a small number of PLA foci was observed, these are specific, as they are sensitive to various controls, such as DRB or TPL treatments or KD of NSUN2. We also check the antibodies background staining in single antibody controls (Figure 1a).

To support our conclusion and PLA data, we have now performed zC-ChIP-seq using NSUN2 antibody and **showing its presence at DSBs genome-wide** (see Figure 2a-e).

Specifically, we treated cells with 5-aza-C, which has been described previously (Hussain et al., Cell Reports, 2013). 5-aza-Cytosine (5-aza-C) is a cytosine analogue that can trap RNA methyltransferases like NSUN2 by forming an irreversible covalent complex with the enzyme.

NSUN2 normally catalyses the methylation of cytosine at the carbon-5 (C5) position in RNA to form 5-methylcytosine (m⁵C), a modification important for RNA stability, translation, and stress responses. This process involves the enzyme recognizing its RNA substrate, where a catalytic cysteine in NSUN2 performs a nucleophilic attack on the C6 position of the cytosine ring, forming a covalent intermediate. A methyl group from S-adenosylmethionine (SAM) is then transferred to the C5 position of cytosine, after which the enzyme is released. However, when 5-aza-C is incorporated into RNA in place of cytosine, its nitrogen atom at the C5 position prevents the methyl group from being transferred, blocking the completion of the methylation reaction. NSUN2 still forms the initial covalent intermediate with 5-aza-C at the C6 position, but because the methylation step cannot proceed, the enzyme becomes irreversibly trapped on the RNA. This mechanism is exploited experimentally to identify RNA targets of NSUN2 by "trapping" the enzyme at its binding sites. As DARTs are in close proximity to chromatin, this experimental approach further strengthens our conclusion that NSUN2 **binds and methylates** DARTS.

Finally, as suggested, we have performed NSUN2/DARTs FISH-PLA and show the overlap with gH2AX IF (Figure 3d).

2. The laser track quality is poor. On a laser track similar to this image, many SSBs and gaps, which are repaired by BER or NER, are generated. The author should show that the laser condition primarily induces DSBs, or that it contains SSBs and gaps as well as DSBs. In addition, the gH2AX or 53BP1 signal obtained by IF or live imaging should be shown at the laser track. GFP alone should also be shown because such high-power lasers can recruit GFP fragments. As suggested, we have repeated NSUN2-Neon-green laser stripes and show Neon-green alone as well (which is negative). This is now shown in Figure 1b. The same settings were used in our previous publication, showing 53BP1 and TIRR recruitment to DSBs (Ketley et al, Cell Reports, 2022, Figure 5H).

3. Why did the authors use TPL in Fig. 1 and DRB in Fig. 4? In general, because transcription inhibitors block transcription with different mechanisms, multiple transcription inhibitors are tested. Both TPL and DRB should be tested, at least in Fig 1 and Fig. 4

As suggested, we have repeated experiments in Figure 1a and added both transcription inhibitors and show the same outcome: loss of NSUN2/gH2AX foci.

Similarly, we have now included both DRB and TLP also for NSUN2/Dicer PLA experiments, now shown in new Figure 5b and Extended Data Figure 10a.

4. In Fig. 1b, why did the authors not examine the left side of the AsiSI site? Transcription inhibitors should also be tested.

We appreciate this comment. To address this fully, we have performed zC-NSUN2-ChIP-seq to show its presence around DSBs genome wide. These new data are added to new Figure 2a-e.

5. The authors preferred to use a PLA assay to demonstrate the interaction in this study. PLA is a tool used to detect protein-protein(or nucleotides) interactions under a microscope; however, non-specific signals are often detected by PLA. The authors should confirm these interactions using other methods such as a co-IP. In addition, for example, NGS analysis in combination with co-IP can also be applied by using the AsiSI system.

We appreciate this comment. Although our PLA experiments usually contain several controls, such as treatments with RNAses, transcription inhibitors, or knock downs of relevant proteins, as well as single antibody controls, it is useful to have other experimental approaches to strengthen the conclusions.

As suggested, we have now performed zC-NSUN2-ChIP-seq to show its presence around DSBs genome wide. These new data are added to new Figure 2a-e.

6. The authors have mentioned that BLESS was performed in the manuscript (Fig. 3a). A heatmap of BLESS should be included, because siNSUN2 may change the distribution of DSB induction. In addition, the authors should present the results of the sense/antisense reads of the -4OHT sample.

In the current version, the scale of the x-axis is +/- 2.5 kbp. the authors should show a wider area because the RNA signal may spread over 2.5 kbp from the AsiSI site in siNSUN2 cells.

We apologise for the confusion, we have not performed BLESS, this was done and published by Prof Legube's lab, who also invented this now commonly used system for sequence specific DSB induction. We used their BLESS data in our previous publications and here as an indication of cleaved AsiSI sites (there are around 1200 sites in the genome, but only about 100 are always cleaved).

To address this comment, we have performed gH2AX IF in AsiSI system in control and siNSUN2 cells and show that the total level of gH2AX is not changing. These data are now added to the Extended Data Figure 8b.

As suggested, we also show new heatmaps, showing -4OHT data in a bigger x-axis window (Extended Data Figure 7b).

7. Fig. 3. f–g: the result for the low transcription DSB site should be shown.

Data for low transcription are shown in Extended Data Figure 8c.

8. In Figs 4–6, whether this is a DSB-dependent interaction is unclear. Only a few PLA foci were induced after irradiation with 10 Gy or approximately 300 DSBs. This interaction should be confirmed using a PLA-independent method.

We also show Surface plasmon resonance (Figure 5a) to support the PLA data.

9. In Fig. 7, the authors performed the comet, cell survival, and DR-GFP assays in RAD51- or NSUN2-depleted cells. The magnitude of repair defect in siNSUN2 cells is very minor although it is statistically significant. In particular, the defect in the DR-GFP assay is very small. Therefore, whether these results provide evidence that NSUN2 promotes HR is unclear. To further confirm this data, RAD51 foci per cell should be examined. In addition, double knockdown of siNSUN2+siRAD51 should be performed to address whether it is epistatic. The cell cycle profile should be shown because a low percentage of the S/G2 phase affects HR efficiency in the DR-GFP assay. Why did the authors use BRCA1 siRNA in DR-GFP? The siRAD51 should be used.

As suggested, we have now repeated HR reporter system in cells depleted of RAD51 as a control (instead of BRCA1). We also included double knock down of NSUN2 and RAD51 in this experiment. These new data are now shown in Figure 7c.

We also tested the cell cycle profiles in cells depleted of NSUN2 and show no significant change, see Extended Data Figure 16b.

10. Time course should be performed to examine DSB repair defect using gH2AX foci analysis. The initial induction of gH2AX foci formation may be changed in the NSUN2 mutant.

We have performed time course to examine gH2AX foci at 1h, 4h and 24h.

The initial induction of gH2AX at 1h post IR was the same in all samples. Please see Figure 7d and Extended Data Figure 16d.

11. The use of the word of “DART” by the authors is confusing because the concept of DNA damage-induced RNA is reported by Fabrizio d’Adda di Fagagna as damage-induced long non-coding RNAs (dilncRNAs).

We agree with the reviewer that there are several similarities between DARTs and dilncRNA. In fact, we believe that primary DARTs, which start from the break as de novo transcripts, are the same as dilncRNA. The reason for calling them DARTs is that dilncRNA are proposed to serve as a substrate for DDRNA generation. However, we have not detected DDRNA at unique DSBs.

Other labs, such as Visa, have also only detected them at DSBs in repetitive regions and not at unique DSBs. Furthermore, we showed that R-loops at the end of primary DARTs can serve as promoters for secondary DARTs, which are de novo transcripts coming back towards the breaks. In contrast, dilncRNA were shown to be single transcripts firing from the broken ends.

Reviewer #3 (Remarks to the Author):

The study investigates the molecular mechanisms of NSUN2, an RNA m5C-methyltransferase involved in the repair of double-stranded breaks (DSB). It was found that NSUN2 localizes to DSBs in a transcription-dependent manner, and binds to and methylates DARTs, which are DNA damage response RNAs. Furthermore, they detected an RNA-dependent interaction between NSUN2 and DICER, which was stimulated by DNA damage. NSUN2 activity was shown to promote DICER cleavage of DARTs-associated R-loops, essential for efficient DNA repair. **The topics of the study are interesting.** However, I found that the study did not convincingly demonstrate the working models of NSUN2. Below are my specific comments.

We are very grateful for this reviewer's comments. We hope that the revision of the manuscript now contains data that provide a convincing model of NSUN2 at DSBs.

1. In Figure 1a, the Proximity Ligation Assay (PLA) included a single antibody control for NSUN2 under -IR conditions and for γ H2AX under +IR conditions. Ideally, they should have included single antibody controls for both proteins under both -IR and +IR conditions to ensure comprehensive control data. Additionally, the plot lacks clear indications of what the y-axis represents, and it is not specified what each dot in the plot signifies.

To address this comment, we now show single antibody controls for both proteins, NSUN2 and γ H2AX, in both conditions, -IR and +IR. We also indicate on y-axis that this corresponds to number of PLA foci per nucleus. Each dot represents a cell with certain number of PLA foci. We added more explanation to the figure legend.

2. In Figure 1b, the authors introduced TPL into the experiment without providing an explanation for its inclusion. Further clarification is needed on why TPL was used in this context. Additionally, the correctness of the "Relative Fluorescence Signal" equation mentioned in the figure legend should be confirmed for accuracy.

First of all, we have now repeated laser stripes and include GFP alone control, see new Figure 1b. We have also corrected the figure legend, thank you for pointing this out.

TPL was added as a transcription inhibitor which requires only 1 hour of treatment and is irreversible. In Figure 1, we were testing whether transcription is required for the recruitment of NSUN2 to DSBs. Indeed, TPL treatment led to inhibition of NSUN2 recruitment to DSBs.

3. Moreover, the authors need to describe in more detail the differences in the fluorescence signal in the cell images before and after damage. Specifically, an explanation of how the fluorescence signal disappears upon laser treatment would aid in understanding the dynamics involved. Furthermore, to substantiate the role of TPL in this experiment, it is essential to include an examination that confirms whether TPL treatment results in transcription inhibition effectively.

To address this comment, we have revised the manuscript to provide a more comprehensive explanation of the changes in fluorescence signal before and after laser stripe-induced damage. Specifically, we write that laser irradiation can cause general photobleaching of the fluorescence signal throughout the entire cell, which contributes to the observed decrease in signal intensity. In addition to photobleaching, we discuss the possible dynamic redistribution of fluorescently labelled proteins at the site of damage.

To justify the role of triptolide (TPL) in our experiment, we have included a reference to previous studies demonstrating that TPL effectively inhibits transcription by targeting the XPB subunit of TFIIH, thereby blocking RNA polymerase II-dependent transcription initiation (Titov et al., 2011, Nature Chemical Biology; Vispé et al., 2009, Molecular Cancer Therapeutics). These findings provide strong evidence that TPL treatment results in transcriptional inhibition, supporting its use in our study to assess transcription-dependent responses to DNA damage.

4. In Figure 1c, the authors examined the localization of NSUN2 at the DSB using the DiVA system. However, they only investigated the presence of NSUN2 at the DS1 locus. To provide a more comprehensive analysis, it would be beneficial for the authors to conduct sequencing to examine the localization of NSUN2 on a genome-wide scale and verify this localization across several loci.

To fully address this comment, we have performed zC-NSUN2-ChIP-seq and show its localisation at DSBs genome wide. Please see new Figure 2a-e and below for more details.

5. In Figure 2, the authors utilized zC-FISH-PLA to examine the interaction of NSUN2 with DARTs at two loci, DS2 and DS3. The results indicated that the NSUN2 protein interacted with antisense DARTs at DS2 but not with DS3 itself. It is unclear why the authors chose to conduct these assays using DS2 and DS3 instead of DS1. Additionally, NSUN2 was only found at DS3, prompting concerns about whether the zC-FISH-PLA technique is sensitive enough to detect interactions between NSUN2 and DARTs, or if NSUN2 can only interact with certain DARTs. Given that the interaction was confirmed at only one locus (DS2), it is premature to conclude generally that NSUN2 binds to antisense DARTs based on this single instance. Further investigation across more loci and with enhanced sensitivity in detection methods would help in establishing a more robust conclusion.

5-aza-Cytosine (5-aza-C) is a cytosine analog that can trap RNA methyltransferases like NSUN2 by forming an irreversible covalent complex with the enzyme. NSUN2 normally catalyzes the methylation of cytosine at the carbon-5 (C5) position in RNA to form 5-methylcytosine (m⁵C), a modification important for RNA stability, translation, and stress responses. This process involves the enzyme recognizing its RNA substrate, where a catalytic cysteine in NSUN2 performs a nucleophilic attack on the C6 position of the cytosine ring, forming a covalent intermediate. A methyl group from S-adenosylmethionine (SAM) is then transferred to the C5 position of cytosine, after which the enzyme is released. However, when 5-aza-C is incorporated into RNA in place of cytosine, its nitrogen atom at the C5 position prevents the methyl group from being transferred, blocking the completion of the methylation reaction. NSUN2 still forms the initial covalent intermediate with 5-aza-C at the C6 position, but because the methylation step cannot proceed, the enzyme becomes irreversibly trapped on the RNA. This mechanism is exploited experimentally to identify RNA targets of NSUN2 by "trapping" the enzyme at its binding sites. As DARTs are in close proximity to chromatin, this experimental approach further strengthens our conclusion that NSUN2 binds and methylates DARTs. Furthermore, we include pioneering data showing Direct RNA sequencing using Oxford Nanopore Technology, showing NSUN2 dependent methylation on DARTs.

Direct RNA sequencing using Oxford Nanopore Technology (ONT) enables the detection of RNA modifications like 5-methylcytosine (m⁵C) by analyzing the characteristic changes they induce in the ionic current as RNA strands pass through the nanopore. Unlike traditional sequencing methods that require reverse transcription or amplification, ONT reads native RNA molecules directly, preserving modification information. Modified bases like m⁵C alter the electrical signal differently compared to unmodified cytosine, allowing trained machine learning models to identify and distinguish these modifications with increasing accuracy. Furthermore, we have validated these data in siNSUN2 samples.

See Figure 2 and Extended Figures 2, 3 and 4. We hope that all these new data now provide robust evidence for NSUN2 activity at DSBs, genome wide.

6. The authors stated that the formation of the zC-RNA-NSUN2 complex signifies the active addition of m⁵C to RNA transcripts. To substantiate this claim, they should provide additional details on how this process occurs.

Please see the explanation in the previous point and new data. This information has been added to the text.

7. In Figure 2b, it is necessary to include a Western blot analysis to verify the NSUN2 protein levels after knockdown (KD).

We have validated and optimised the level of NSUN2 knock down many times. For representative image, please see Western blot in Extended Data Figure 16a.

8. It is essential to provide evidence confirming that NSUN2 has been effectively knocked down and that 4-hydroxytamoxifen (4-OHT) is functional. This could include data from Western blot analyses to demonstrate the reduction in NSUN2 protein levels post-knockdown, and functional assays or reporter gene activity to verify the efficacy of 4-OHT.

We have validated and optimised the level of NSUN2 knock down many times. For representative image, please see Western blot in Extended Data Figure 16a.

The functionality of 4-OHT was monitored by the induction of gH2AX (Extended Data Figure 8b).

9. Why was there a sudden switch to using DRB instead of continuing with TPL? It would be helpful for the authors to explain the rationale behind this change in the choice of inhibitors. To be honest there was no strong reason for switching, my lab is using both all the time. We now provide data showing both inhibitors in one assay, please see Figure 1a and Figure 5b.

10. In Figures 6a and b, where it is claimed that NSUN2 facilitates DICER processing of DART-associated R-loops, this conclusion should be carefully reevaluated. Firstly, to ensure the purity of the used DICER enzyme and to exclude any nuclease contamination, it is recommended that the authors conduct cleavage assays using some pre-miRNAs as quality control experiments. We appreciate this comment. Indeed, we do observe additional bands on the Coomassie stain of the purified recombinant Dicer. However, these samples were further concentrated and purified with a CYTIVA Vivaspin, removing any contaminants smaller than 100kDa. We have added additional data to Extended Data Figure 9c. After this step only one major contaminant can be observed and this was identified as Dicer fragment by Mass Spectroscopy. Indeed, such fragment has been shown by others, for example Camino et al., Mol Cell 2023, Figure S6. Furthermore, if indeed there still would be any nucleases present, we would detect unspecific cleavage of the RNA substrates even in Dicer alone sample. This was not observed, please see Figure 6a, Dicer (no NSUN2) samples. Therefore, we conclude that our Dicer cleavage assay detects Dicer enzymatic activity.

Secondly, the methodology involving qPCR to examine the original RNA and the remaining RNA after cleavage using three different pairs of primers appears to yield inconsistent results. Therefore, the interpretation of these results should be reconsidered. The authors could also run the RNA on a gel to directly determine the cleavage efficiency of DICER, which would provide a more direct method for quantification. These additional experiments and clarifications would strengthen the validity of the claim regarding NSUN2's role in facilitating DICER processing of R-loops.

Initially, we indeed performed Dicer cleavage assay with urea PAGE gel readout. However, this proved to be very difficult to quantify due to various exposure times, see Extended Data Figure 12. This is similar to the western blots for protein analysis, which are considered as only semi-quantitative. Therefore, we turn to quantitative qPCR readout.

We apologise, but we do not fully understand what this reviewer means by "inconsistent results". We used four different RT primers, three within the R-loop and one just outside to monitor closely Dicer cleavage of the R-loop substrate. In all cases we see only minor activity of Dicer alone on this RNA substrate. Dicer cleavage activity is enhanced by the presence of NSUN2 wt in all four RT-qPCR reactions, and less so in presence of NSUN2 K190M. Dicer cleavage is also enhanced on pre-methylated R-loop tail substrate.

Additionally, we are now providing further controls for this reaction. Specifically, we show that indeed our substrate is a bona fide R-loop, (it is cleaved by RNaseH) see Extended Data Figure 15c. We also have repeated Dicer cleavage assay using in vitro methylated R-loop vs non-methylated R-loop to show that Dicer prefers methylated R-loops, see Figure 6a.

Furthermore, we now show additional in vivo data. Specifically, we isolated R-loops from wt and Dicer KD cells in absence or presence of NSUN2 and DNMT2 (as a control) and subjected them to S9.6 slot blot. We show now that depletion of Dicer and NSUN2 leads to accumulation of R-loops in vivo upon irradiation. Please see Figure 6c.

11. In Figure 6d, it is necessary to include at least three experimental repeats to ensure the robustness and reproducibility of the results.

We have now replaced these data with more specific slot blot, showing S9.6 levels in wt and Dicer kd in absence or presence on NSUN2 and DNMT2 upon DNA damage induction. We show these data in triplicates as requested. See new Figure 6c.

Furthermore, the m5C slot blot is also now shown in triplicates in Extended Data Figure 15a.

REVIEWER COMMENTS

Reviewer #1 (Remarks to the Author):

The authors have done a good job in addressing my previous concerns. These new data now strengthen the claims and conclusions of the manuscript, and I recommend publication of this study in Nature Communications.

We are very grateful to this reviewer for all helpful and constructive suggestions that improved our manuscript and for their recommendation for publication in Nature Communications.

Reviewer #2 (Remarks to the Author):

The authors did not properly understand my original request regarding the γ H2AX immunofluorescence. As I clearly stated in my previous review, 10 Gy induces approximately 300–600 DSBs per nucleus. At 15 minutes post-irradiation, the majority of these breaks are still unrepaired. Consequently, staining with γ H2AX at this time point results in pan-nuclear signal distribution, meaning that virtually any nuclear protein may appear to colocalize with γ H2AX simply due to signal saturation.

To address this concern, we performed PLA using HOXD11, a chromatin-associated transcription factor not known to play a role at double-strand breaks and used as a negative control in our previous study (Liu et al., EMBO J, 2024) together with γ H2AX antibodies, in both control and 10 Gy-irradiated cells. No significant increase in PLA foci (at background levels) was observed upon DNA damage induction, confirming that the PLA assay does **not** produce false positives due to pan-nuclear γ H2AX signal. These data are now included in Extended Data Figure 1b. Additionally, we show expression levels of HOXD11 and γ H2AX by immunofluorescence in Extended Data Figure 1c.

Overall, our NSUN2/ γ H2AX PLA experiment includes following controls:

1. +/- IR to show DNA damage specificity
2. DRB and TPL inhibitors to show transcription dependency
3. NSUN2 and γ H2AX single antibody controls for specificity of the signal
4. NSUN2 KD to show specificity of the signal
5. control negative samples with HOXD11 to exclude the possibility of pan-nuclear false positive signal
6. IR dose dependency to support physiological and specific signal

We hope that all these controls now satisfy this reviewer's concerns about NSUN2/ γ H2AX PLA.

To illustrate this point, the authors should provide a representative γ H2AX IF image at 15 minutes post-10 Gy. Furthermore, because Figure 1a shows 20–60 PLA signals per nucleus, a dose-response experiment (e.g., 2, 5, 10 Gy=10, 30, 60 PLA signal should be detected) should be performed to demonstrate specificity and dynamic range. Importantly, the colocalization of γ H2AX foci with PLA signals must be shown using samples post-IR, not AsiSI-induced breaks. As requested, we performed a dose titration experiment with irradiation at 0, 2, 5, and 10 Gy. PLA was conducted using NSUN2 and γ H2AX antibodies, and co-localisation was assessed via immunofluorescence. We observed a dose-dependent increase in both γ H2AX and NSUN2/ γ H2AX PLA foci. These results confirm specificity and dynamic range of the assay, and are presented in Extended Data Figure 1a.

Again, the authors did not address my request adequately regarding the laser track system. As I previously pointed out, the laser irradiation induces not only DSBs but also SSBs and base

damage. Therefore, the authors must validate whether this laser system predominantly induces DSBs, rather than other types of damage. This could be achieved by testing the recruitment of SSB repair or BER factors. Although NSUN2 recruitment to the AsiSI site was shown, the CPM increase is very modest. If NSUN2 is mainly recruited to SSBs or base damage rather than DSBs in the laser track system, the conclusions drawn from these experiments could be misleading.

We completely agree with this reviewer's point that laser striping induces a combination of DSBs, SSBs, and base damage. However, we respectfully disagree that demonstrating the recruitment of an SSB repair factor to laser tracks would clarify NSUN2's specificity for DSBs. Such an approach would not provide direct evidence regarding NSUN2's recruitment to DSBs.

We employed a well-validated laser striping protocol previously used in multiple publications to demonstrate DSB repair factor recruitment. For instance, we showed recruitment of 53BP1 and TIRR in Ketley et al., Cell Reports 2022 (Fig. 5), hSSB1 in Long et al., Cell Reports 2023 (Fig. 1), and INTS6 in Long et al., NAR 2024 (Fig. 1). The same laser parameters were used in this study to assess NSUN2 localisation.

Recognising the limitations in specificity of laser-based damage induction, we did not rely on it exclusively. To robustly determine whether NSUN2 is indeed recruited to DSBs, we used **two additional independent approaches**: (1) site-specific enzymatic induction of DSBs using the AsiSI system followed by ChIP-seq, and (2) well controlled PLA following 10 Gy irradiation, using γ H2AX as a DSB marker, see above. Across these three orthogonal systems, NSUN2 consistently localised to sites of DSBs.

Furthermore, 5-aza-C ChIP-seq data show NSUN2 recruitment to DSBs induced by endonuclease cleavage. NSUN2 is an RNA-binding protein whose chromatin association is RNA-dependent, unlike canonical DNA- or chromatin-binding proteins such as histones. We employed an optimised 5-aza-mediated double crosslinking protocol designed to trap RNA-associated proteins on chromatin. Although the observed CPM might seem modest, it is **highly statistically significant ($p = 1.3e-14$)** and consistent with the expected signal range for RNA-binding proteins.

In Figure 3, there is no textual description of the results involving Exonuclease VII or RNase H in the Results section. More importantly, the observed induction of the PLA signal is marginal, making it difficult to evaluate whether RNase H treatment significantly attenuates the PLA signal. In fact, the RNase H-treated sample post-4OHT still shows approximately 3–5 PLA foci, raising serious concerns about the specificity of the observed signal reduction.

The addition of Exonuclease VII and RNase H1 was performed as requested by Reviewer #1. Exonuclease VII was used to confirm that the FISH-PLA probe is not binding to post-resection ssDNA overhangs. RNase H1 treatment was included to assess whether NSUN2 binds to DSB-derived transcripts (DARTs) that form R-loops. This information has now been added to the manuscript text.

While the induction of FISH-PLA signal may appear modest, it is important to emphasise that this is not a standard PLA assay. FISH-PLA detects the proximity of a protein to a nascent transcript, which is then processed. Due to the nature of this interaction and the transient, low-abundance nature of these RNA intermediates, only a limited number of foci are expected to be detected. Nevertheless, the FISH-PLA signal is **clearly responsive to multiple controls** and is **significantly increased under damage condition**, as well as it is reduced following triptolide or RNase H1 treatments, demonstrating its specificity.

Additionally, it should be noted that DARTs can exist in single-stranded, double-stranded and, R-loop configurations. RNase H1 specifically degrades RNA within RNA–DNA hybrids, and thus only eliminates the subset of DARTs present in R-loops. Other transcript forms remain detectable, which explains the residual PLA signal following RNase H1 treatment.

Overall, the data presented in Figure 3 demonstrate that NSUN2 binds to DSB-derived transcripts (DARTs). This conclusion is further supported by our genome-wide NSUN2 5-aza-C CHIP-seq analysis, in which NSUN2 is covalently trapped on its RNA substrate, confirming its RNA-dependent chromatin association.

Regarding Figures 3d and 5e, it is unclear whether the presented images are projections or single z-slices. The authors should provide representative monolayer images or 3D reconstructions to verify spatial relationships between PLA and γ H2AX foci.

As requested, we provide single plane images in Figures 3d and 5e. This information has been added to the corresponding figure legends.

The magnitude of the DNA repair defects observed in the HR and NHEJ assays, as well as in the γ H2AX assay, appears to be marginal. It remains unclear whether these small differences have biological significance.

DR-GFP and EJ5-GFP reporter cell lines are widely used tools to study homologous recombination and non-homologous end joining repair pathways, respectively. These systems rely on the induction of a single, site-specific double-strand break (DSB) by the I-SceI endonuclease and measure repair efficiency based on the restoration of GFP expression. However, a major limitation of these assays is that only a small fraction of cells ultimately express GFP, reflecting successful repair through the intended pathway. This low percentage arises from multiple factors, including the efficiency of I-SceI cutting and the cell cycle stage of the cells. Moreover, because these reporters are based on artificial substrates integrated at non-endogenous loci, they do not fully recapitulate the chromatin environment or complexity of physiological DNA damage. As a result, depletion of repair factors often leads to only mild phenotypes in these systems. This is further compounded by the limited sensitivity of a single-cut reporter to detect subtle defects. Consequently, while useful for mechanistic insights, DR-GFP and EJ5-GFP assays may underestimate the functional impact of certain DNA repair proteins, especially under conditions that more accurately reflect endogenous damage and cellular stress.

Therefore, in addition, we performed comet assay, which is better than reporter systems for detecting a DNA repair phenotype, because it directly measured DNA damage repair. As shown in Figure 7a, there is a **robust and **** significant phenotype** in cells depleted of RAD51 (control) and NSUN2.

The number of γ H2AX foci should be quantified, rather than relying solely on intensity measurements. In addition, although the 4 h and 24 h images in the control condition are relatively dark, visible foci remain, suggesting the authors may have overinterpreted the results. As requested, we now provide quantification of γ H2AX foci in Extended Data Figure 16e. The conclusion remains the same.

Lastly, for Extended Figure 2h, box plots for additional transcriptionally active and inactive sites should be provided to support the generality of the findings.

As requested, these are now provided in Extended Figure 2e and f.

Reviewer #3 (Remarks to the Author):

I have no further comments. The responses from the authors are good. Thank you.

We are very grateful to this reviewer for all helpful and constructive suggestions that made our manuscript better and acceptable for publication in Nature Communications.